# Transfer Learning via Minimizing the Performance Gap Between Domains

**Boyu Wang**
Department of Computer Science
University of Western Ontario
bwang@csd.uwo.ca

**Jorge A. Mendez**
Department of Computer and Information Science
University of Pennsylvania
mendezme@seas.upenn.edu

**Ming Bo Cai**
Princeton Neuroscience Insititute
Princeton University
mcai@princeton.edu

**Eric Eaton**
Department of Computer and Information Science
University of Pennsylvania
eeaton@seas.upenn.edu

## Abstract

We propose a new principle for transfer learning, based on a straightforward intuition: if two domains are similar to each other, the model trained on one domain should also perform well on the other domain, and vice versa. To formalize this intuition, we define the *performance gap* as a measure of the discrepancy between the source and target domains. We derive generalization bounds for the instance weighting approach to transfer learning, showing that the performance gap can be viewed as an algorithm-dependent regularizer, which controls the model complexity. Our theoretical analysis provides new insight into transfer learning and motivates a set of general, principled rules for designing new instance weighting schemes for transfer learning. These rules lead to gapBoost, a novel and principled boosting approach for transfer learning. Our experimental evaluation on benchmark data sets shows that gapBoost significantly outperforms previous boosting-based transfer learning algorithms.

## 1 Introduction

Transfer learning is based on the idea that learning a new concept is easier after having learned one or more similar concepts. By extracting knowledge from a set of related concepts (source domains) and then leveraging this knowledge upon learning the concept of interest (target domain), the learning performance can be improved. This is especially beneficial when there is insufficient data to learn solely from the target domain, but enough knowledge from the source domains is available. Transfer learning has become increasingly relevant over the last two decades, and consequently during that time various algorithms have been proposed [10, 17, 39, 22, 24, 11], accompanied by theoretical and empirical justifications [4, 25, 3, 21, 18, 19, 26].

In order to successfully transfer information from one domain to another, it is critical to understand the similarities and differences between the domains. Intuitively, the more similar the two domains are, the more information can be transferred. When the domains are considerably different, but still related, a common strategy to correct this difference is to minimize some measure of divergence between the empirical source and target data distributions. Most prior work in this area has focused on defining discrepancy measures that motivate the design of algorithms that effectively reduce the dissimilarity between domains as much as possible [16, 17, 35, 6, 2, 34, 3, 25, 7, 14, 1, 33]. These works have mainly considered the problem of *domain adaptation*, where examples from the target

domain are entirely unlabeled. However, in many practical cases, there is a small amount of labeled target data, which can be leveraged to derive more specialized measures of domain divergence.

To address this issue, we present the first analysis for instance weighting transfer learning that considers the presence of labeled target examples. The contribution of our work is two-fold. **1.** We address the question of **how to measure the divergence between two domains given label information for the target domain.** Intuitively, if two domains are similar to each other, the model trained on one domain should also perform well on the other domain, and vice versa. To formalize this intuition, we propose the notion of *performance gap* between the source and target domains, and show that the transfer learning model complexity can be upper bounded in terms of this performance gap. In other words, it can be viewed as an algorithm-dependent regularizer, which leads to finer and more informative generalization bounds. This is, to the best of our knowledge, the first generalization bound for instance-based transfer learning that considers the presence of labeled target data. Moreover, our definition of performance gap is intuitive and generally applicable to any form of transfer. Thus, our analysis provides a deeper understanding of the general problem of transfer learning and new insight into how to leverage the labeled target examples. **2.** On the algorithmic side, instead of directly minimizing the generalization bound, which is highly computationally expensive, we propose four principled rules to follow when designing an instance weighting scheme for transfer learning. We instantiate these rules with `gapBoost`, a novel and efficient boosting algorithm for transfer learning, which offers out-of-the-box usability and readily accommodates any algorithm for transfer learning. Source code for `gapBoost` is available at `https://github.com/bwang-ml/gapBoost`.

## 2   Related Work

The large majority of transfer learning techniques can be categorized as instance, feature, or parameter transfer [29, 40, 7]. In this paper, we consider the instance transfer approach, where the objective is to correct the difference between the domains by weighting the instances. In this context, the authors in [5, 3] studied transfer learning algorithms that minimize a convex combination of the source and target empirical risks, and proposed to use the $\mathcal{H}$-*divergence* [4] to measure the distance between the domains for 0-1 loss classification. This study was generalized to arbitrary loss functions by introducing the notion of *discrepancy distance* [25]. Since then, various measures have been proposed in the literature [35, 6, 2, 34, 7, 14, 33]. Recently, instance weighting has been revisited in [21] based on the notion of algorithmic stability. The authors revealed that the source domain features can be interpreted as a regularization matrix, which benefits the learning process of the target domain task.

Despite the wide applicability of the discrepancy measures defined in these works, they fail to address three problems. **1.** These measures are designed for the setting of *domain adaptation*, where no label information is available in the target domain. As a result, it is unclear how to leverage any labeling information from the target domain in cases where it is available. Moreover, deriving generalization bounds for domain adaptation requires additional assumptions. One common assumption is that there exists an ideal hypothesis that performs well on both domains [3, 25], which cannot be empirically verified due to the lack of labeled examples in the target domain. **2.** These measures are either algorithm-independent [16, 35, 6, 2, 34] or defined over a hypothesis class [3, 25, 7, 14, 33], and so they ignore the specific algorithm used. An algorithm-specific notion of divergence measure could lead to more informative generalization guarantees. **3.** From the algorithmic perspective, most methods are restricted to linear hypotheses (or nonlinear hypotheses defined through a reproducing kernel Hilbert space) and derive the instance weights by directly minimizing the generalization bounds or divergence measures, which usually imposes a high computational burden [25, 14, 7].

Having access to labeled examples in the target domain enables us to derive more efficient learning algorithms. In [10], an efficient transfer boosting method was proposed to reweight the data for classification in the presence of labeled target data. Later, this approach was extended to regression [30] and multi-source transfer [41, 12]. In [36], the authors proposed a two-stage instance weighting approach for transfer learning, and analyzed its generalization bound by extending the result from [3]. While these algorithms are effective in practice, no theoretical results have been presented to show why transfer learning succeeds when labeled information from the target domain is available.

In addition, most existing theoretical studies of transfer learning examine the convergence rate of the Rademacher complexity or stability coefficient, assuming that the model complexity (and hence the loss function) of the transfer learning algorithm is upper bounded by a constant. One related

work we are aware of that relaxes this assumption is [42], which proves the boundedness of the loss functions in the setting of multitask learning. However, their analysis only relates the bounds with regularization functions and requires the additional assumption that when the hypothesis outputs 0, the loss function is upper bounded by another constant. Critically, these analyses do not provide any insight into how the domain divergence affects the model complexity and the generalization bound. More recently, this issue has been studied in [38], showing that the model complexity of parameter-sharing multitask learning algorithms is determined by the task similarities. However, this theoretical result has not motivated any concrete algorithm.

In contrast to prior work, we derive an algorithm-specific generalization bound that considers the label information from the target domain. Based on our newly developed theory, we design a principled and efficient instance weighting transfer learning algorithm.

## 3 Instance Weighting for Transfer Learning

In this section, we formalize the problem of instance weighting for transfer learning. We continue by proposing four general rules to follow when developing new weighting schemes, along with the theoretical grounds that support these rules. We then instantiate these new rules with gapBoost.

Let $z = (x, y) \in \mathcal{X} \times \mathcal{Y}$ be a training example drawn from some unknown distribution $\mathcal{D}$, where $x$ is the data point, and $y$ is its label, with $\mathcal{Y} = \{-1, 1\}$ for binary classification and $\mathcal{Y} \subseteq \mathbb{R}$ for regression. A hypothesis is a function $h \in \mathcal{H}$ that maps $\mathcal{X}$ to the set $\mathcal{Y}'$ sometimes different from $\mathcal{Y}$, where $\mathcal{H}$ is a hypothesis class. For a convex, non-negative loss function $\ell : \mathcal{Y}' \times \mathcal{Y} \mapsto \mathbb{R}_+$, we denote by $\ell(h(x), y)$ the loss of hypothesis $h$ at point $z = (x, y)$. Let $S = \{z_i = (x_i, y_i)\}_{i=1}^N$ be a set of $N$ training examples drawn independently from $\mathcal{D}$. The empirical loss of $h$ on $S$ and its generalization loss over $\mathcal{D}$ are defined, respectively, by $\mathcal{L}_S(h) = \frac{1}{N}\sum_{i=1}^N \ell(h(x_i), y_i)$, and $\mathcal{L}_{\mathcal{D}}(h) = \mathbb{E}_{z \sim \mathcal{D}}[\ell(h(x), y)]$. We consider the linear function class in a Euclidean space, but our analysis is also applicable to a reproducing kernel Hilbert space. We also assume that $\|x\|_2 \leq R, \forall x \in \mathcal{X}$ for some $R \in \mathbb{R}_+$, and the loss function is $\rho$-Lipschitz continuous for some $\rho \in \mathbb{R}_+$.

In the setting of transfer learning, we have a training sample $S = \{S_{\mathcal{T}}, S_{\mathcal{S}}\}$ of size $N = N_{\mathcal{T}} + N_{\mathcal{S}}$ composed of $S_{\mathcal{T}} = \{z_i^{\mathcal{T}} = (x_i^{\mathcal{T}}, y_i^{\mathcal{T}})\}_{i=1}^{N_{\mathcal{T}}}$ drawn from a target distribution $\mathcal{D}_{\mathcal{T}}$ and $S_{\mathcal{S}} = \{z_i^{\mathcal{S}} = (x_i^{\mathcal{S}}, y_i^{\mathcal{S}})\}_{i=1}^{N_{\mathcal{S}}}$ drawn from a source distribution $\mathcal{D}_{\mathcal{S}}$. We analyze the transfer learning algorithms based on instance weighting, which aims to optimize the following objective function:

$$\min_{h \in \mathcal{H}} \mathcal{L}_S^{\Gamma}(h) + \lambda \mathcal{R}(h) \ , \tag{1}$$

where $\mathcal{L}_S^{\Gamma}(h) = \mathcal{L}_{S_{\mathcal{T}}}^{\Gamma^{\mathcal{T}}}(h) + \mathcal{L}_{S_{\mathcal{S}}}^{\Gamma^{\mathcal{S}}}(h)$ is the *weighted* empirical loss over the source and target domains, $\mathcal{R}(h)$ is a regularization function to control the model complexity of $h$, and $\lambda$ is a regularization parameter. The domain-specific weighted losses are given by $\mathcal{L}_{S_{\mathcal{T}}}^{\Gamma^{\mathcal{T}}}(h) = \sum_{i=1}^{N_{\mathcal{T}}} \gamma_i^{\mathcal{T}} \ell(h(x_i^{\mathcal{T}}), y_i^{\mathcal{T}})$ and $\mathcal{L}_{S_{\mathcal{S}}}^{\Gamma^{\mathcal{S}}}(h) = \sum_{i=1}^{N_{\mathcal{S}}} \gamma_i^{\mathcal{S}} \ell(h(x_i^{\mathcal{S}}), y_i^{\mathcal{S}})$. The instance weights $\Gamma = [\Gamma^{\mathcal{T}}; \Gamma^{\mathcal{S}}]$, with $\Gamma^{\mathcal{T}} = [\gamma_1^{\mathcal{T}}, \ldots, \gamma_{N_{\mathcal{T}}}^{\mathcal{T}}]^\top \in \mathbb{R}_+^{N_{\mathcal{T}}}$ and $\Gamma^{\mathcal{S}} = [\gamma_1^{\mathcal{S}}, \ldots, \gamma_{N_{\mathcal{S}}}^{\mathcal{S}}]^\top \in \mathbb{R}_+^{N_{\mathcal{S}}}$, are such that $\sum_{i=1}^{N_{\mathcal{T}}} \gamma_i^{\mathcal{T}} + \sum_{i=1}^{N_{\mathcal{S}}} \gamma_i^{\mathcal{S}} = 1$, and they can either be learned in a pre-processing step [17, 8, 28, 15] or incorporated into learning algorithms [10, 23]. As we consider the linear function class, the hypothesis $h$ has the form of an inner product $h(x) = \langle h, x \rangle$, and we study the regularization function $\mathcal{R}(h) = \|h\|_2^2$.

### 3.1 Principles for Instance Weighting

Leveraging problem (1) requires assigning appropriate values to $\Gamma$ so that the solution to problem (1) leads to effective transfer. There are a variety of weighting schemes developed in the literature. In this paper, we summarize four general and intuitively reasonable rules as follows. As we will show later, they are also theoretically grounded.

1. Minimize the weighted empirical loss over source and target domains, as suggested by (1).

2. Assign balanced weights to data points, as focusing too much on specific data points leads to overfitting caused by perturbations in the training data [32].

3. Assign more weight to the target sample, since target data will be used for testing.

4. Assign weights such that the *performance gap* between the domains is small.

Our main contribution lies in Rule 4, for which we introduce the notion of performance gap. Although intuitive, these rules are contradictory, so designing an algorithm based on them requires properly trading them off. We explore one way to control this tradeoff via hyper-parameters in Section 3.3.

## 3.2 Theoretical Justifications

We now develop the theoretical foundations that justify the instance weighting rules. In contrast to previous studies on domain adaptation, we propose a notion to measure the divergence between the domains that leverages the label information, leading to Rule 4 in our instance weighting scheme. Intuitively, *if two domains are similar, the model trained on one domain should also perform well on the other.* To make this intuition precise, we define the notion of *performance gap* below.

**Definition 1** (**Performance gap**). *Let* $\mathcal{V}_{\mathcal{S}}(h) = \mathcal{L}_{S_{\mathcal{S}}}^{\Gamma^{\mathcal{S}}}(h) + \eta\lambda\mathcal{R}(h)$ *and* $\mathcal{V}_{\mathcal{T}}(h) = \mathcal{L}_{S_{\mathcal{T}}}^{\Gamma^{\mathcal{T}}}(h) + \eta\lambda\mathcal{R}(h)$, *respectively, be the objective functions in the source and target domains, where* $\eta \in (0, \frac{1}{2})$, *and let their minimizers, respectively, be* $h_{S_{\mathcal{S}}}$ *and* $h_{S_{\mathcal{T}}}$. *The performance gap between the source and target domains is defined as*

$$\nabla = \nabla_{\mathcal{T}} + \nabla_{\mathcal{S}},$$

*where* $\nabla_{\mathcal{S}} = \mathcal{L}_{S_{\mathcal{S}}}^{\Gamma^{\mathcal{S}}}(h_{S_{\mathcal{T}}}) - \mathcal{L}_{S_{\mathcal{S}}}^{\Gamma^{\mathcal{S}}}(h_{S_{\mathcal{S}}})$ *and* $\nabla_{\mathcal{T}} = \mathcal{L}_{S_{\mathcal{T}}}^{\Gamma^{\mathcal{T}}}(h_{S_{\mathcal{S}}}) - \mathcal{L}_{S_{\mathcal{T}}}^{\Gamma^{\mathcal{T}}}(h_{S_{\mathcal{T}}})$.

Note that the performance gap is both data and algorithm dependent, which is crucial for deriving a more informative and finer generalization bound. Moreover, note that, although we use the performance gap to analyze the specific setting of instance weighting, it could be readily applied to other transfer learning paradigms, such as feature-based transfer. We now present the definition of $\mathcal{Y}$-Discrepancy, which we require for our analysis.

**Definition 2** ($\mathcal{Y}$-**Discrepancy** [27]). *Let* $\mathcal{H}$ *be a hypothesis class mapping* $\mathcal{X}$ *to* $\mathcal{Y}$ *and let* $\ell : \mathcal{Y} \times \mathcal{Y} \mapsto \mathbb{R}_+$ *define a loss function over* $\mathcal{Y}$. *The* $\mathcal{Y}$-*discrepancy distance between two distributions* $\mathcal{D}_1$ *and* $\mathcal{D}_2$ *over* $\mathcal{X} \times \mathcal{Y}$ *is defined as:*

$$\mathrm{dist}_{\mathcal{Y}}(\mathcal{D}_1, \mathcal{D}_2) = \sup_{h \in \mathcal{H}} |\mathcal{L}_{\mathcal{D}_1}(h) - \mathcal{L}_{\mathcal{D}_2}(h)| \ .$$

Our main theoretical contribution is the following theorem that bounds the difference between $\mathcal{L}_{\mathcal{D}_{\mathcal{T}}}$ and $\mathcal{L}_S^{\Gamma}$, which justifies our principles for instance weighting.

**Theorem 1.** *Let* $h_S$ *be the optimal solution of the transfer learning problem (1). Assume that* $\|x\|_2 \leq R, \forall x \in \mathcal{X}$, *and that the loss function is* $\rho$-*Lipschitz continuous and convex. Then, for any* $\delta \in (0, 1)$, *with probability at least* $1 - \delta$, *we have*

$$\mathcal{L}_{\mathcal{D}_{\mathcal{T}}}(h_S) \leq \mathcal{L}_S^{\Gamma}(h_S) + \varepsilon_{\Gamma} + \|\Gamma^{\mathcal{S}}\|_1 \,\mathrm{dist}_{\mathcal{Y}}(\mathcal{D}_{\mathcal{T}}, \mathcal{D}_{\mathcal{S}}) \ , \tag{2}$$

*where*

$$\varepsilon_{\Gamma} = \min\left\{ \frac{\|\Gamma\|_{\infty}\rho^2 R^2}{\lambda} + \left( \frac{\rho^2 R^2(\|\Gamma\|_2^2 + \|\Gamma\|_{\infty})}{\lambda} + \|\Gamma\|_{\infty}B(\Gamma) \right)\sqrt{\frac{N\log\frac{1}{\delta}}{2}}, \right.$$

$$\left. \frac{2\|\Gamma\|_{\infty}\|\Gamma\|_2\rho^2 R^2}{\lambda}\sqrt{2N\log\frac{4}{\delta}} + \|\Gamma\|_2 B(\Gamma)\sqrt{\frac{\log\frac{2}{\delta}}{2}} \right\}.$$

**Remark 1.** *Rule 1 is justified by* $\mathcal{L}_S^{\Gamma}$, *Rule 2 is justified by* $\|\Gamma\|_2$ *and* $\|\Gamma\|_{\infty}$, *and Rule 3 is justified by* $\|\Gamma^{\mathcal{S}}\|_1$. $B(\Gamma)$ *is an upper bound of the loss function* $\ell$, *such that* $\ell(h(x), y) \leq B(\Gamma)$, *where* $h$ *is the output hypothesis of an algorithm solving the transfer learning problem (1). We emphasize that it is a function of* $\Gamma$ *and, as we show later, can be upper bounded in terms of* $\nabla$, *which justifies Rule 4.*

**Proof Sketch.** (Details of the proof are available in the appendix)

**Step 1: Bound** $\mathcal{L}_{\mathcal{D}_{\mathcal{T}}}$ **from** $\mathcal{L}_{\mathcal{D}}^{\Gamma}$. Let $\mathcal{L}_{\mathcal{D}}^{\Gamma} = \mathcal{L}_{\mathcal{D}_{\mathcal{T}}}^{\Gamma^{\mathcal{T}}} + \mathcal{L}_{\mathcal{D}_{\mathcal{S}}}^{\Gamma^{\mathcal{S}}}$ be the expected weighted loss of $\mathcal{L}_S^{\Gamma}$. Then, by linearity of the expectation and the definition of $\mathcal{Y}$-discrepancy, we show that the following holds:

$$\mathcal{L}_{\mathcal{D}_{\mathcal{T}}} \leq \mathcal{L}_{\mathcal{D}}^{\Gamma} + \|\Gamma^{\mathcal{S}}\|_1 \,\mathrm{dist}_{\mathcal{Y}}(\mathcal{D}_{\mathcal{T}}, \mathcal{D}_{\mathcal{S}}) \ . \tag{3}$$

**Remark 2.** *Compared to the notion of discrepancy [25], one advantage of $\mathcal{Y}$-discrepancy is that it does not require the assumption that the loss function obeys the triangle inequality [3, 9], which does not hold for many loss functions (e.g., hinge loss, squared loss), to make (3) hold. In addition, we can prove that for a binary classification problem, $\mathrm{dist}_{\mathcal{Y}}(\mathcal{D}_{\mathcal{T}}, \mathcal{D}_{\mathcal{S}})$ can be upper bounded from a finite sample by constructing a new classification problem, where the positive target examples and negative source examples are positively labeled, and the negative target examples and positive source examples are negatively labeled. See Lemma A and Lemma B in the appendix for more details.*

**Step 2: Bound $\mathcal{L}_{\mathcal{D}}^{\Gamma}$ from $\mathcal{L}_{S}^{\Gamma}$.** We present two schemas to upper bound $\mathcal{L}_{\mathcal{D}}^{\Gamma}$: one is based on algorithmic stability, and the other one is based on Rademacher complexity, which lead to the definition of $\varepsilon_{\Gamma}$.

**Algorithmic stability bound.** We introduce the notion of *weight-dependent uniform stability* (see Definition A in the appendix) and show that, for any $\delta \in (0, 1)$, with probability at least $1 - \delta$, the expected loss $\mathcal{L}_{\mathcal{D}_{\mathcal{T}}}$ can be upper bounded by:

$$\mathcal{L}_{\mathcal{D}}^{\Gamma} \leq \mathcal{L}_{S}^{\Gamma} + \frac{\|\Gamma\|_{\infty}\rho^2 R^2}{\lambda} + \left( \frac{\rho^2 R^2(\|\Gamma\|_2^2 + \|\Gamma\|_{\infty})}{\lambda} + \|\Gamma\|_{\infty}B(\Gamma) \right) \sqrt{\frac{N \log \frac{1}{\delta}}{2}} \ . \tag{4}$$

**Rademacher complexity bound.** We introduce the notion of *weighted Rademacher complexity* (see Definition B in the appendix ), and relate it to the notion of *uniform argument stability* [20]. Then, we prove that the learning algorithm (1) produces an *algorithmic hypothesis class $\mathcal{B}$*, and, for any $\delta \in (0, 1)$, with probability at least $1 - \delta$, the expected loss $\mathcal{L}_{\mathcal{D}_{\mathcal{T}}}$ can be upper bounded by:

$$\mathcal{L}_{\mathcal{D}}^{\Gamma} \leq \mathcal{L}_{S}^{\Gamma} + 2\frac{\|\Gamma\|_{\infty}\|\Gamma\|_2\rho^2 R^2}{\lambda} \sqrt{2N \log \frac{4}{\delta}} + B(\Gamma)\|\Gamma\|_2\sqrt{\frac{\log \frac{2}{\delta}}{2}} \ . \tag{5}$$

Combining (3), (4), and (5), we obtain the generalization bound (2).

**Remark 3.** *If $\gamma_i = \frac{1}{N}, \forall i \in \{1, \ldots, N\}$, we recover the standard argument stability bound from (2), which suggests assigning equal weights to all instances to achieve a fast convergence rate, due to $\|\Gamma\|_{\infty}$ and $\|\Gamma\|_2$. In particular, if $\|\Gamma\|_{\infty}$ (and hence $\|\Gamma\|_2^2$) is $\mathcal{O}(\frac{1}{N})$, (2) leads to a convergence rate of $\mathcal{O}(\frac{1}{\sqrt{N}})$. However, in the setting of transfer learning, it is usually the case that $N_{\mathcal{T}} \ll N_{\mathcal{S}}$. Consequently, we may have $\|\Gamma\|_{\infty} \ll \frac{1}{N_{\mathcal{T}}}$, which implies that transfer learning has a faster convergence rate than single-task learning. On the other hand, as we will show in Step 3, the loss bound $B$ is also a function of $\Gamma$, which suggests a new criterion for instance weighting.*

**Step 3: Bound $B(\Gamma)$.** The following lemma shows that the model complexity of the transfer learning algorithm (1) can be upper bounded in terms of the performance $\nabla$.

**Lemma 1.** *Let $h_S$ be the optimal solution of the instance weighting transfer learning problem (1). Then, we have:*

$$\|h_S\|_2 \leq \sqrt{\frac{\nabla}{2\lambda(1 - 2\eta)} + \frac{\|h_{S_{\mathcal{S}}}\|_2^2 + \|h_{S_{\mathcal{T}}}\|_2^2}{2}} \ .$$

By bounding the model complexity, we obtain various upper bounds for different loss functions.

**Corollary 1.** *The hinge loss function of the learning algorithm (1) can be upper bounded by:*

$$B(\Gamma) \leq 1 + R\sqrt{\frac{\nabla}{2\lambda(1 - 2\eta)} + \frac{\|h_{\mathcal{S}}\|_2^2 + \|h_{\mathcal{T}}\|_2^2}{2}} \ .$$

*For regression, if the response variable is bounded by $|y| \leq Y$, the $\ell_q$ loss of (1) can be bounded by:*

$$B(\Gamma) \leq \left( Y + R\sqrt{\frac{\nabla}{2\lambda(1 - 2\eta)} + \frac{\|h_{\mathcal{S}}\|_2^2 + \|h_{\mathcal{T}}\|_2^2}{2}} \right)^q \ .$$

**Algorithm 1 gapBoost**

---

**Input:** $S_\mathcal{S}, S_\mathcal{T}, K, \rho_\mathcal{S} \leq \rho_\mathcal{T} \leq 0, \gamma_{\max}$, a learning algorithm $\mathcal{A}$

1: Initialize $D_1^\mathcal{S}(i) = D_1^\mathcal{T}(i) = \frac{1}{N_\mathcal{S}+N_\mathcal{T}}$ for all $i$

2: **for** $k = 1, \ldots, K$ **do**

3:     Call $\mathcal{A}$ to train a base learner $h_k$ using $S_\mathcal{S} \cup S_\mathcal{T}$ with distribution $D_k^\mathcal{S} \cup D_k^\mathcal{T}$

4:     Call $\mathcal{A}$ to train an auxiliary learner $h_k^\mathcal{S}$ over source domain using $S_\mathcal{S}$ with distribution $D_k^\mathcal{S}$

5:     Call $\mathcal{A}$ to train an auxiliary learner $h_k^\mathcal{T}$ over target domain using $S_\mathcal{T}$ with distribution $D_k^\mathcal{T}$

6:     $\epsilon_k = \sum\limits_{i=1}^{N_\mathcal{S}} D_k^\mathcal{S}(i) \mathbb{1}_{h_k(x_i^\mathcal{S}) \neq y_i^\mathcal{S}} + \sum\limits_{i=1}^{N_\mathcal{T}} D_k^\mathcal{T}(i) \mathbb{1}_{h_k(x_i^\mathcal{T}) \neq y_i^\mathcal{T}}, \ \alpha_k = \log \frac{1-\epsilon_k}{\epsilon_k}$

7:     **for** $i = 1, \ldots, N_\mathcal{S}$ **do**

8:         $\beta_i^\mathcal{S} = \rho_\mathcal{S} \mathbb{1}_{h_k^\mathcal{S}(x_i^\mathcal{S}) \neq h_k^\mathcal{T}(x_i^\mathcal{S})} + \alpha_k \mathbb{1}_{h_k(x_i^\mathcal{S}) \neq y_i^\mathcal{S}}, \ D_{k+1}^\mathcal{S}(i) = D_k^\mathcal{S}(i) \exp\left(\beta_i^\mathcal{S}\right)$

9:     **end for**

10:     **for** $i = 1, \ldots, N_\mathcal{T}$ **do**

11:         $\beta_i^\mathcal{T} = \rho_\mathcal{T} \mathbb{1}_{h_k^\mathcal{S}(x_i^\mathcal{T}) \neq h_k^\mathcal{T}(x_i^\mathcal{T})} + \alpha_k \mathbb{1}_{h_k(x_i^\mathcal{T}) \neq y_i^\mathcal{T}}, \ D_{k+1}^\mathcal{T}(i) = D_k^\mathcal{T}(i) \exp\left(\beta_i^\mathcal{T}\right)$

12:     **end for**

13:     $Z_{k+1} = \sum_{i=1}^{N_\mathcal{S}} D_{k+1}^\mathcal{S}(i) + \sum_{i=1}^{N_\mathcal{T}} D_{k+1}^\mathcal{T}(i)$

14:     **if** $D_{k+1}^\mathcal{S}(i), D_{k+1}^\mathcal{T}(i) > \gamma_{\max} Z_{k+1}$ **then**

15:         $D_{k+1}^\mathcal{S}(i), D_{k+1}^\mathcal{T}(i) = \gamma_{\max} Z_{k+1}$

16:     **end if**

17:     Normalize $D_{k+1}^\mathcal{S}$ and $D_{k+1}^\mathcal{T}$ such that $\sum_{i=1}^{N_\mathcal{S}} D_{k+1}^\mathcal{S}(i) + \sum_{i=1}^{N_\mathcal{T}} D_{k+1}^\mathcal{T}(i) = 1$

18: **end for**

**Output:** $f(x) = \text{sign}\left(\sum_{k=1}^{K} \alpha_k h_k(x)\right)$

---

**Remark 4.** *Lemma 1 shows that, given fixed weights, the model complexity (and hence the upper bound of a loss function) is related to the performance gap between the source and target domains. Lemma 1 reveals that transfer learning (1) can succeed when the hypotheses trained on their own domains also work well on the other domains, which leads to a lower training loss and a faster convergence to the best hypothesis in the class in terms of sample complexity.*

By combining the Steps 1–3, we obtain Theorem 1. $\qquad\square$

By similar derivations, we obtain a PAC learning bound, which is also consistent with the instance weighting rules.

**Corollary 2.** *Let $w_S$ be the optimal solution of the transfer learning problem (1), and $h^* = \arg\min_h \mathcal{L}_{\mathcal{D}_\mathcal{T}}(h)$ be the minimizer in the target domain. Assume that $\|x\|_2 \leq R, \forall x \in \mathcal{X}$, and that the loss function obeys the triangle inequality and is $\rho$-Lipschitz and convex. Then, for any $\delta \in (0,1)$, with probability at least $1 - \delta$, we have:*

$$\mathcal{L}_{\mathcal{D}_\mathcal{T}}(h_S) \leq \mathcal{L}_{\mathcal{D}_\mathcal{T}}(h^*) + \varepsilon_\Gamma' + 2\|\Gamma^\mathcal{S}\|_1 \, \text{dist}_\mathcal{Y}(\mathcal{D}_\mathcal{T}, \mathcal{D}_\mathcal{S}) \ , \tag{6}$$

*where*

$$\varepsilon_\Gamma' = \min \left\{ \frac{\|\Gamma\|_\infty \rho^2 R^2}{\lambda} + \left( \frac{\rho^2 R^2 (\|\Gamma\|_2^2 + \|\Gamma\|_\infty)}{\lambda} + \left( \frac{\|\Gamma\|_2}{\sqrt{N}} + \|\Gamma\|_\infty \right) B(\Gamma) \right) \sqrt{\frac{N \log \frac{4}{\delta}}{2}}, \right.$$

$$\left. \frac{2\|\Gamma\|_\infty \|\Gamma\|_2 \rho^2 R^2}{\lambda} \sqrt{2N \log \frac{8}{\delta}} + 2\|\Gamma\|_2 B(\Gamma) \sqrt{\frac{\log \frac{4}{\delta}}{2}} \right\}.$$

### 3.3 gapBoost

As $\text{dist}_\mathcal{Y}(\mathcal{D}_\mathcal{T}, \mathcal{D}_\mathcal{S})$ can be estimated from the training sample, it is possible to derive a weighting scheme by minimizing the generalization bounds (2) as in previous works in the literature [25, 7]. However, one common issue with this approach is that it leads to high computational cost for large sample size and it is usually restricted to linear hypotheses. In contrast, our algorithmic goal is to derive a computationally efficient method that is applicable to large-scale data and also flexible enough to accommodate arbitrary learning algorithms for transfer learning.

Table 1: Comparison of boosting algorithms for transfer learning.

| | Rule 1 | Rule 2 | Rule 3 | Rule 4 |
|---|---|---|---|---|
| AdaBoost | ✓ | ✓ | ✗ | ✗ |
| TrAdaBoost | ✓ | ✗ | ✓ | ✗ |
| TransferBoost | ✓ | ✗ | ✓ | ✗ |
| gapBoost | ✓ | ✓ | ✓ | ✓ |

To this end, we propose gapBoost in Algorithm 1, which explicitly exploits the rules from Section 3.1. The algorithm trains a joint learner for source and target domains, as well as auxiliary source and target learners (lines 3–5). Then, it up-weights incorrectly labeled instances as per traditional boosting methods and down-weights instances for which the source and target learners disagree; the trade-off for the two schemes is controlled separately for source and target instances via hyper-parameters $\rho_{\mathcal{S}}$ and $\rho_{\mathcal{T}}$ (lines 6–12). Finally, the weights are clipped to a maximum value of $\gamma_{\max}$ and normalized (lines 13–17). **1.** gapBoost follows Rule 1 by training the base learner $h_k$ at each iteration, which aims to minimize the weighted empirical loss over the source and target domains. **2.** By tuning $\gamma_{\max}$, it explicitly controls $\|\Gamma\|_{\infty}$ and implicitly controls $\|\Gamma\|_2$, as required by Rule 2. Additionally, as each base learner $h_k$ is trained with a different set of weights, the final classifier $f$ returned by gapBoost is potentially trained over a balanced distribution. **3.** Moreover, by setting $\rho_{\mathcal{T}} \geq \rho_{\mathcal{S}}$, gapBoost penalizes instances from the source domain more than from the target domain, implicitly assigning more weight to the target domain sample than to the source domain sample, as suggested by Rule 3. **4.** Finally, as $\rho_{\mathcal{S}}, \rho_{\mathcal{T}} \leq 0$, the weight of any instance $x$ will decrease if the learners disagree (i.e., $h_k^{\mathcal{S}}(x) \neq h_k^{\mathcal{T}}(x)$). By doing so, gapBoost follows Rule 4 by minimizing the gap $\nabla$. **5.** The trade-off between the rules is balanced by the choice of the hyper-parameters $\rho_{\mathcal{T}}$, $\rho_{\mathcal{S}}$ and $\gamma_{\max}$.

Table 1 compares various traditional boosting algorithms for transfer learning in terms of the instance weighting rules. Conventional AdaBoost [13] treats source and target samples equally, and therefore does not reduce $\|\Gamma^{\mathcal{S}}\|_1$ or minimize the performance gap. On the other hand, TrAdaBoost [10] and TransferBoost [12] explicitly exploit Rule 3 by assigning less weight to the source domain sample at each iteration. However, they do not control $\|\Gamma\|_{\infty}$ or $\|\Gamma\|_2$, so the weight of the target domain sample can be large after a few iterations. Most critically, none of the previous algorithms minimize the performance gap explicitly as we do, which can be crucial for transfer learning to succeed.

The generalization performance of gapBoost is upper bounded by the following proposition.

**Proposition 1.** *Let $f(x) = \sum_{k=1}^K \alpha_k h_k(x)$ be the ensemble of classifiers returned by* gapBoost, *with each base learner trained by solving (1). For simplicity, we assume that $\sum_{k=1}^K \alpha_k = 1$. Then, for any $\delta \in (0,1)$, with probability at least $1 - \delta$, we have*

$$\mathcal{L}_{\mathcal{D}_{\mathcal{T}}}(f) \leq \mathcal{L}_{S_{\mathcal{T}}}(f) + \frac{2\rho^2 R^2 \gamma_{\infty}^{\mathcal{T}}}{\lambda} \sqrt{2 \log \frac{4}{\delta}} + B(\Gamma) \sqrt{\frac{\log \frac{2}{\delta}}{2 N_{\mathcal{T}}}} \ .$$

*where $\gamma_{\infty}^{\mathcal{T}}$ is the largest weight of the target sample over all boosting iterations.*

**Remark 5.** *We observe that if $\gamma_{\infty}^{\mathcal{T}} \gg \sqrt{\frac{1}{N_{\mathcal{T}}}}$, the bound will be dominated by the second term. Then, Proposition 1 suggests to set $\gamma_{max} = \mathcal{O}(\frac{1}{\sqrt{N_{\mathcal{T}}}})$ to achieve a fast convergence rate. On the other hand, as the loss function is convex, $B(\Gamma)$ can be upper bounded by $B(\Gamma) \leq \sum_{k=1}^K \alpha_k B(\Gamma_k)$, where $\Gamma_k$ is the set of weights at the $k$-th boosting iteration. In other words, one should aim to minimize the performance gap for every boosting iteration to achieve a tighter bound.*

## 4   Experiments

We evaluated gapBoost on two benchmark data sets.

*20 Newsgroups* This data set contains approximately 20,000 documents, grouped by seven top categories and 20 subcategories. Each transfer learning task involved a top-level classification problem, while the source and target domains were chosen from different subcategories. The source and target data sets were in the same way as in [10], yielding 6 transfer learning problems.

*Office-Caltech* This data set contains approximately 2,500 images from four distinct domains: Amazon (A), DSLR (D), Webcam (W), and Caltech (C), which enabled us to construct 12 transfer

Table 2: Comparison of different methods on the 20 Newsgroups (top) and Office-Caltech (bottom) data sets in term of error rate (%). The row titles are standard names used in the literature to identify the transfer problems. Our algorithm, gapBoost, outperforms all baselines in the majority of transfer problems, and is competitive with the top performance in the remaining ones. Standard error is reported after the $\pm$.

| | AdaBoost$_{\mathcal{T}}$ | AdaBoost$_{\mathcal{T}\&\mathcal{S}}$ | TrAdaBoost | TransferBoost | gapBoost |
|---|---|---|---|---|---|
| comp vs sci | $12.45 \pm 0.47$ | $13.45 \pm 0.48$ | $12.03 \pm 0.41$ | $8.83 \pm 0.37$ | $\mathbf{7.68 \pm 0.25}$ |
| rec vs sci | $10.99 \pm 0.37$ | $11.79 \pm 0.35$ | $10.03 \pm 0.36$ | $7.93 \pm 0.30$ | $\mathbf{7.39 \pm 0.21}$ |
| comp vs talk | $11.83 \pm 0.42$ | $14.57 \pm 0.47$ | $10.67 \pm 0.37$ | $\mathbf{6.45 \pm 0.25}$ | $7.10 \pm 0.27$ |
| comp vs rec | $15.80 \pm 0.53$ | $17.50 \pm 0.64$ | $14.86 \pm 0.67$ | $12.11 \pm 0.43$ | $\mathbf{9.81 \pm 0.29}$ |
| rec vs talk | $12.08 \pm 0.36$ | $9.40 \pm 0.31$ | $12.21 \pm 0.40$ | $6.26 \pm 0.30$ | $\mathbf{5.66 \pm 0.21}$ |
| sci vs talk | $11.74 \pm 0.49$ | $10.52 \pm 0.37$ | $10.13 \pm 0.46$ | $6.45 \pm 0.26$ | $\mathbf{5.92 \pm 0.24}$ |
| A $\rightarrow$ C | $43.87 \pm 0.52$ | $27.76 \pm 0.88$ | $37.57 \pm 0.68$ | $27.86 \pm 0.82$ | $\mathbf{27.06 \pm 0.87}$ |
| A $\rightarrow$ D | $32.65 \pm 1.35$ | $28.33 \pm 1.33$ | $34.93 \pm 1.43$ | $28.96 \pm 1.38$ | $\mathbf{25.08 \pm 1.37}$ |
| A $\rightarrow$ W | $37.23 \pm 0.98$ | $26.94 \pm 1.17$ | $31.03 \pm 0.95$ | $26.95 \pm 1.15$ | $\mathbf{24.34 \pm 1.10}$ |
| C $\rightarrow$ A | $39.92 \pm 0.74$ | $20.32 \pm 0.80$ | $29.13 \pm 0.80$ | $19.68 \pm 0.80$ | $\mathbf{19.13 \pm 0.83}$ |
| C $\rightarrow$ D | $27.88 \pm 1.14$ | $25.69 \pm 1.19$ | $\mathbf{19.84 \pm 1.09}$ | $23.44 \pm 1.33$ | $21.03 \pm 1.20$ |
| C $\rightarrow$ W | $30.25 \pm 1.05$ | $24.50 \pm 1.30$ | $22.86 \pm 0.95$ | $23.41 \pm 1.30$ | $\mathbf{21.55 \pm 1.20}$ |
| D $\rightarrow$ A | $44.30 \pm 0.45$ | $40.86 \pm 0.39$ | $45.33 \pm 0.48$ | $\mathbf{40.50 \pm 0.44}$ | $40.66 \pm 0.39$ |
| D $\rightarrow$ C | $44.00 \pm 0.56$ | $40.09 \pm 0.46$ | $43.72 \pm 0.62$ | $40.35 \pm 0.46$ | $\mathbf{40.00 \pm 0.46}$ |
| D $\rightarrow$ W | $50.63 \pm 0.58$ | $49.64 \pm 0.66$ | $49.95 \pm 0.65$ | $\mathbf{49.63 \pm 0.65}$ | $50.24 \pm 0.62$ |
| W $\rightarrow$ A | $42.91 \pm 0.46$ | $37.22 \pm 0.56$ | $44.24 \pm 0.52$ | $\mathbf{37.02 \pm 0.53}$ | $37.04 \pm 0.52$ |
| W $\rightarrow$ C | $44.12 \pm 0.50$ | $37.93 \pm 0.58$ | $44.78 \pm 0.65$ | $37.79 \pm 0.56$ | $\mathbf{37.48 \pm 0.50}$ |
| W $\rightarrow$ D | $40.63 \pm 1.45$ | $45.52 \pm 1.58$ | $\mathbf{40.00 \pm 1.51}$ | $44.88 \pm 1.58$ | $41.74 \pm 1.40$ |

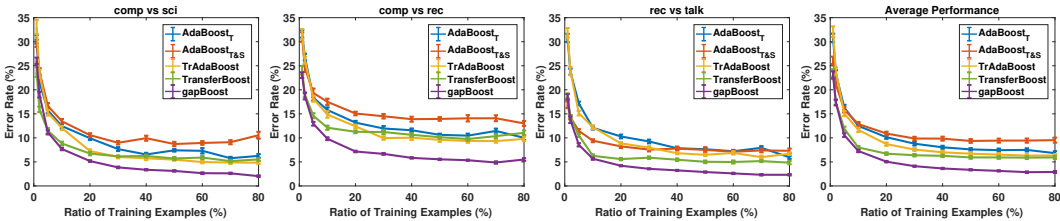

Figure 1: Test error rates (%) with different sizes of target sample on different tasks and on average across all tasks. gapBoost consistently outperforms the baselines on all regimes of target sample size. Since gapBoost more effectively leverages the target instances, its improvement over the baselines is more noticeable as the target sample size increases. Error bars represent standard error.

problems by alternately selecting each possible source-target pair. All four domains share the same 10 classes, so we constructed 5 binary classification tasks for each transfer problem and the averaged results are reported.

**Performance comparison** We evaluated gapBoost against four baseline algorithms: AdaBoost$_{\mathcal{T}}$ trained only on target data, AdaBoost$_{\mathcal{T}\&\mathcal{S}}$ trained on both source and target data, TrAdaBoost, and TransferBoost. Logistic regression is used as the base learner for all methods, and the number of boosting iterations is set to 20. The hyper-parameters of gapBoost were set as $\gamma_{\max} = \frac{1}{\sqrt{N_{\mathcal{T}}}}$ as per Remark 5, $\rho_{\mathcal{T}} = 0$, which corresponds to no punishment for the target data, and $\rho_{\mathcal{S}} = \log \frac{1}{2}$.

In both data sets we pre-processed the data using principal component analysis (PCA) to reduce the the feature dimension to 100. For each data set, we used all source data and a small amount of target data (10% on 20 Newsgroups and 10 points on Office-Caltech) as training sample, and used the rest of the target data for testing. We repeated all experiments over 20 different random train/test splits and the average results are presented in Table 2, showing that our method is capable of outperforming all the baselines in the majority of cases. In particular, gapBoost consistently outperforms AdaBoost$_{\mathcal{T}}$, empirically indicating that it avoids *negative transfer*.

**Learning with different number of target examples** To further investigate the effectiveness of gapBoost, we varied the fraction of target instances of the 20 Newsgroups data set used for training,

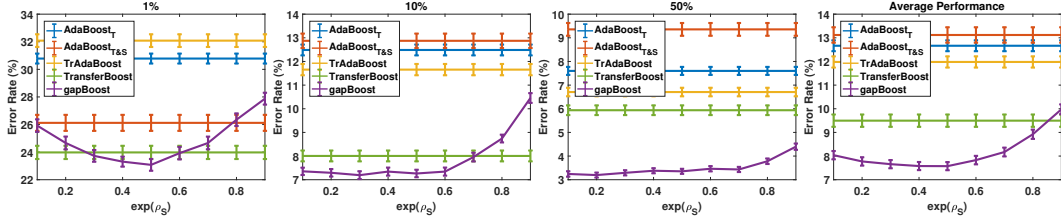

Figure 2: Test error rates (%) averaged across all tasks with respect to the values of the hyper-parameter $\rho_{\mathcal{S}}$ for varying sample sizes. Rightmost graphic shows results averaged over all sample sizes. gapBoost becomes less sensitive to the choice of $\rho_{\mathcal{S}}$ as the target sample grows larger. In all cases, there is a range of $\rho_{\mathcal{S}}$ that outperforms all baselines. Error bars represent standard error.

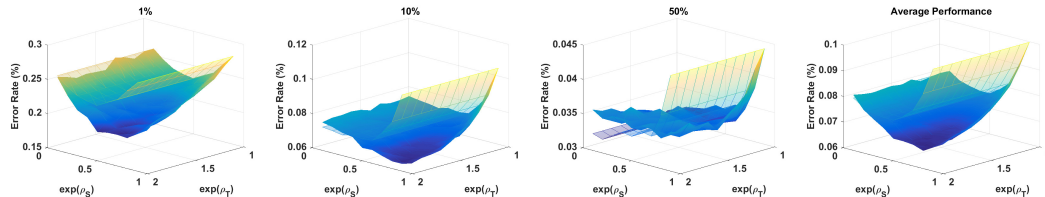

Figure 3: Test error rates (%) with varying $\rho_{\mathcal{S}}$ and $\rho_{\mathcal{T}}$. The valley curves correspond to $\rho_{\mathcal{T}} = 0$ (i.e., the purple curves in Figure 2). Hence, regions below the curve indicate better hyper-parameters.

from 0.01 to 0.8. Figure 1 shows full learning curves on three example tasks, as well as the average performance over all six tasks. The results reveal that gapBoost's improvement over the baselines increases as the number of target instances grows, indicating that it is able to leverage target data more effectively than previous methods.

**Parameter sensitivity**    Next, we empirically evaluated our algorithm's sensitivity to the choice of hyper-parameters. We first fixed $\rho_{\mathcal{T}} = 0$ and varied $\exp(\rho_{\mathcal{S}})$ in the range of $[0.1, \ldots, 0.9]$. Figure 2 shows the results averaged over all transfer problems on the 20 Newsgroups data set, showing that as the size of the target sample increases, the influence of the hyper-parameter on performance decreases. In particular, we see that we are able to obtain a range of hyper-parameters for which our method outperforms all baselines in all sample size regimes.

**Increase the weight of a target instance when** $h_k^{\mathcal{S}}(x_{\mathcal{T}}) = h_k^{\mathcal{T}}(x_{\mathcal{T}})$    To further minimize the gap, we can modify the weight update rule for target data: $\beta^{\mathcal{T}} = \rho_{\mathcal{T}} \mathbb{1}_{h_k^{\mathcal{S}}(x^{\mathcal{T}}) = h_k^{\mathcal{T}}(x^{\mathcal{T}})} + \alpha_k \mathbb{1}_{h_k(x^{\mathcal{T}}) \neq y^{\mathcal{T}}}$ with $\rho_{\mathcal{T}} \geq 0$. We vary $\rho_{\mathcal{S}}$ and $\rho_{\mathcal{T}}$ together, and the results are shown in Figure 3. It can be observed that gapBoost can achieve even better performance by focusing more on performance gap minimization (i.e., choosing large $\rho_{\mathcal{S}}$ and $\rho_{\mathcal{T}}$). As the target data increase, the results are less sensitive to the hyper-parameters.

## 5    Conclusions

We propose the notion of performance gap to measure the divergence between domains in transfer learning by exploiting the label information in the target domain. Consequently, we propose a new principle for transfer learning. In particular, our theoretical analysis justifies four intuitively reasonable rules for instance weighting, and provides new insight into transfer learning. We highlighted the role of performance gap minimization and presented gapBoost, an algorithm that explicitly exploits the rules for instance weighting. The empirical evaluation justifies the effectiveness of our algorithm.

While the theoretical analysis is based on the convexity assumption, our principles are quite general, and so would be applicable to a wide variety of algorithms (such as deep nets) for transfer learning. In addition, the principle of performance gap minimization opens up several avenues for knowledge transfer. For example, it could be used to analyze other forms of transfer learning like parameter or feature transfer [37]. It could also help develop knowledge transfer strategies for other learning paradigms such as meta-learning or lifelong learning [31]. We plan to explore these questions in future work.

## Acknowledgements

The research presented in this paper was supported by the Faculty of Science at the University of Western Ontario and the Lifelong Learning Machines program from DARPA/MTO under grant #FA8750-18-2-0117. We would like to thank the anonymous reviewers for their helpful feedback.

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
