[Supplementary Material]

# Transfer Learning via Minimizing the Performance Gap Between Domains
# Supplementary Materials

**Boyu Wang**
Department of Computer Science
University of Western Ontario
bwang@csd.uwo.ca

**Jorge Mendez**
Department of Computer and Information Science
University of Pennsylvania
mendezme@seas.upenn.edu

**Mingbo Cai**
Princeton Neuroscience Insititute
Princeton University
mcai@princeton.edu

**Eric Eaton**
Department of Computer and Information Science
University of Pennsylvania
eeaton@seas.upenn.edu

This document presents the proofs of the generalization bounds of the instance weighting approach to transfer learning. In Section 1, we present the notions and definitions used throughout the proofs. For the sake of readability and self-containedness, we then recall the theoretical results in Section 2, and their proofs are presented in Section 3. Finally, additional experimental results are reported in Section 4.

## 1 Preliminaries

**Definition 1 (Uniform stability).** *Let $h_S \in \mathcal{H}$ be the hypothesis returned by a learning algorithm $\mathcal{A}$ when trained on sample S. $\mathcal{A}$ is said to have $\beta$-uniform stability, with $\beta \in \mathbb{R}_+$, if the following holds:*

$$\sup_{z \sim \mathcal{D}} |\ell(h_S(x), y) - \ell(h_{S'}(x), y)| \leq \beta \qquad \forall S, S' \ ,$$

*where $S'$ is the training sample S with a single example z replaced by an i.i.d. example $z'$. The smallest such $\beta$ satisfying the inequality is the stability coefficient of $\mathcal{A}$.*

**Definition 2 (Rademacher complexity).** *Let $\mathcal{H}$ be a hypothesis class defined over a set $\mathcal{X}$ drawn from a distribution $\mathcal{D}$, and $S = \{x_i\}_{i=1}^N$ a fixed sample of size N with elements in $\mathcal{X}$. Then, the Rademacher complexity of $\mathcal{H}$ is defined as:*

$$\mathfrak{R}_{\mathcal{D}}(\mathcal{H}) = \mathbb{E} \sup_{h \in \mathcal{H}} \frac{1}{N} \sum_{i=1}^N \sigma_i h(x_i) \ ,$$

*where $\sigma_i, \ldots, \sigma_N$ are are independent uniform random variables taking values in $\{-1, +1\}$.*

**Definition 3 ($\rho$-Lipschitz continuity).** *A loss function $\ell(h(x), y)$ is $\rho$-Lipschitz continuous with respect to the hypothesis class $\mathcal{H}$ for some $\rho \in \mathbb{R}_+$, if, for any two hypotheses $h, h' \in \mathcal{H}$ and for any $(x, y) \in \mathcal{X} \times \mathcal{Y}$, we have:*

$$|\ell(h(x), y) - \ell(h'(x), y)| \leq \rho |h(x) - h'(x)| \ .$$

Another notion used to upper bound the stability coefficient is that of *Bregman divergence* [13].

**Definition 4 (Bregman divergence).** *Let $F : \Omega \to \mathbb{R}$ be a strictly convex function defined on a closed convex set $\Omega$. The Bregman divergence associated with F for $f, g \in \Omega$ is defined as*

$$d_F(f, g) = F(f) - F(g) - \langle f - g, \delta F(g) \rangle \ ,$$

*where $\delta F(g)$ is an arbitrary element of the subgradient of F at g.*

**Definition 5 (Discrepancy [11]).** *Let $\mathcal{H}$ be a hypothesis class mapping $\mathcal{X}$ to $\mathcal{Y}$ and let $\ell : \mathcal{Y} \times \mathcal{Y} \to \mathbb{R}_+$ define a loss function over $\mathcal{Y}$. The discrepancy distance between two distributions $\mathcal{D}_1$ and $\mathcal{D}_2$ over $\mathcal{X}$ is defined by:*

$$\text{dist}(\mathcal{D}_1, \mathcal{D}_2) = \sup_{h,h' \in \mathcal{H}} |\mathcal{L}_{\mathcal{D}_1}(h, h') - \mathcal{L}_{\mathcal{D}_2}(h, h')| \ ,$$

*where we have slightly abused our notation without creating confusion by making $\mathcal{L}_{\mathcal{D}}(h, h') = \mathbb{E}_{x \sim \mathcal{D}}[\ell(h(x), h'(x))]$.*

**Definition 6 ($\mathcal{Y}$-Discrepancy [12]).** *Let $\mathcal{H}$ be a hypothesis class mapping $\mathcal{X}$ to $\mathcal{Y}$ and let $\ell : \mathcal{Y} \times \mathcal{Y} \mapsto \mathbb{R}_+$ define a loss function over $\mathcal{Y}$. The $\mathcal{Y}$-discrepancy distance between two distributions $\mathcal{D}_1$ and $\mathcal{D}_2$ over $\mathcal{X} \times \mathcal{Y}$ is defined as:*

$$\text{dist}_{\mathcal{Y}}(\mathcal{D}_1, \mathcal{D}_2) = \sup_{h \in \mathcal{H}} |\mathcal{L}_{\mathcal{D}_1}(h) - \mathcal{L}_{\mathcal{D}_2}(h)| \ .$$

**Remark 1.** *It is easy to verify that both discrepancy and $\mathcal{Y}$-discrepancy are symmetric and obey the triangle inequality for any loss function, that is, $d(\mathcal{D}_1, \mathcal{D}_2) + d(\mathcal{D}_2, \mathcal{D}_3) \geq d(\mathcal{D}_1, \mathcal{D}_3)$, for $d(\cdot) = \text{dist}(\cdot)$ and $d(\cdot) = \text{dist}_{\mathcal{Y}}(\cdot)$.*

**Definition 7 (Uniform argument stability).** *Let $h_S \in \mathcal{H}$ be the hypothesis returned by a learning algorithm $\mathcal{A}$ when trained on sample $S$ of size $N$. An algorithm $\mathcal{A}$ has $\alpha$-uniform argument stability, with $\alpha \in \mathbb{R}_+$, if the following holds:*

$$\|h_S - h_{S^i}\| \leq \alpha \ , \qquad \forall S, S^i \ \forall i \in \{1, \ldots, N\} \ .$$

**Remark 2.** *Uniform argument stability is a stronger notion than uniform stability. If an algorithm $\mathcal{A}$ has uniform argument stability $\alpha$, and we assume that for all $x \in \mathcal{X}$, $\|x\|_2 \leq R$, then for any $\rho$-Lipschitz continuous loss function, $\mathcal{A}$ has $\beta$-uniform stability with $\beta = \alpha \rho R$.*

The following Lemma shows that the hypothesis output by an argument stable algorithm is concentrated around its expectation $\mathbb{E}h_S$ [7].

**Lemma 1.** *If a learning algorithm $\mathcal{A}$ is an $\alpha$-argument stable algorithm, then, for any $\delta \in (0, 1)$, we have:*

$$Pr\left(\|h - \mathbb{E}h_S\| \leq \alpha\sqrt{2N \log \frac{2}{\delta}}\right) \leq 1 - \delta.$$

Lemma 1 implies that an argument stable algorithm essentially defines a hypothesis class centered at $\mathbb{E}h_S$, rather than the entire hypothesis class $\mathcal{H}$.

**Definition 8 (Algorithmic hypothesis class).** *For a fixed sample of size $N$ and a confidence parameter $\delta \in (0, 1)$, the algorithmic hypothesis class of a stable learning algorithm is defined by:*

$$\mathcal{B} = \left\{ h \in \mathcal{H} \mid \|h - \mathbb{E}h_S\| \leq \alpha\sqrt{2N \log \frac{2}{\delta}} \right\} \ .$$

The following Lemma shows that the Rademacher complexity of $\mathfrak{R}(\mathcal{B})$ of an algorithmic hypothesis class can be upper bounded [7].

**Lemma 2.** *Let $S$ be a training sample of $N$ i.i.d. points drawn from some distribution $\mathcal{D}$, and let $h_S$ be a linear hypothesis that has $\alpha$-uniform argument stability. Then, for any $\delta \in (0, 1)$, with probability at least $1 - \delta$, the weighted Rademacher complexity $\mathfrak{R}_{\mathcal{D}}^{\Gamma}(\mathcal{B})$ can be upper bounded by:*

$$\mathfrak{R}_{\mathcal{D}}(\mathcal{B}) \leq \alpha R\sqrt{2 \log \frac{2}{\delta}} \ .$$

## 2 Theoretical Results

We study the weight-dependent stability of regularized algorithms based on instance weighting. Specifically, given a training set $S = \{z_i = (x_i, y_i)\}_{i=1}^{N}$, and their weights $\Gamma = [\gamma_1, \ldots, \gamma_N]^\top$ such

that $\Gamma \succeq 0$ and $\|\Gamma\|_1 = 1$. We analyze the learning algorithm $\mathcal{A}$ which aims to solve the following objective function:

$$\min_{h \in \mathcal{H}} \mathcal{L}_S^\Gamma(h) + \lambda \mathcal{R}(h) \ , \tag{1}$$

where $\mathcal{L}_S^\Gamma(h) = \sum_{i=1}^N \gamma_i \ell(h(x_i), y_i)$. As we consider the linear function class, the hypothesis $h$ has the form of $h(x) = \langle h, x \rangle$, where $\langle \cdot, \cdot \rangle$ is the inner product of two vectors, and we study the regularization function $\mathcal{R}(h) = \|h\|_2^2$.

**Remark 3.** *A special case of (1) is to minimize a convex combination of the empirical losses of the source and target domains [2, 1, 9]:*

$$\min_{h \in \mathcal{H}} \gamma \mathcal{L}_{S_\mathcal{T}}(h) + (1 - \gamma) \mathcal{L}_{S_\mathcal{S}}(h) + \lambda \mathcal{R}(h) \ ,$$

*where $\gamma \in [0, 1]$ is a weight parameter that controls the tradeoff between target and source domains.*

*Problem (1) also accommodates* domain adaptation *[14], where the only labeled instances are from the source domain[1], giving the following objective function:*

$$\min_{h \in \mathcal{H}} \mathcal{L}_{S_\mathcal{S}}^{\Gamma^\mathcal{S}}(h) + \lambda \mathcal{R}(h) \ .$$

We present our main theoretical results as follows.[2]

**Theorem A.** *Let $h_S$ be the optimal solution of the transfer learning problem (1). Assume that $\|x\|_2 \leq R, \forall x \in \mathcal{X}$, and that the loss function is $\rho$-Lipschitz continuous and convex. Then, for any $\delta \in (0, 1)$, with probability at least $1 - \delta$, we have*

$$\mathcal{L}_{\mathcal{D}_\mathcal{T}}(h_S) \leq \mathcal{L}_S^\Gamma(h_S) + \varepsilon_\Gamma + \|\Gamma^\mathcal{S}\|_1 \operatorname{dist}_{\mathcal{Y}}(\mathcal{D}_\mathcal{T}, \mathcal{D}_\mathcal{S}) \ , \tag{2}$$

*where*

$$\varepsilon_\Gamma = \min \left\{ \frac{\|\Gamma\|_\infty \rho^2 R^2}{\lambda} + \left( \frac{\rho^2 R^2 (\|\Gamma\|_2^2 + \|\Gamma\|_\infty)}{\lambda} + \|\Gamma\|_\infty B(\Gamma) \right) \sqrt{\frac{N \log \frac{1}{\delta}}{2}}, \right.$$

$$\left. \frac{2\|\Gamma\|_\infty \|\Gamma\|_2 \rho^2 R^2}{\lambda} \sqrt{2N \log \frac{4}{\delta}} + \|\Gamma\|_2 B(\Gamma) \sqrt{\frac{\log \frac{2}{\delta}}{2}} \right\}.$$

**Corollary A.** *Let $w_S$ be the optimal solution of the transfer learning problem (1), and $h^* = \arg\min_h \mathcal{L}_{\mathcal{D}_\mathcal{T}}(h)$ be the minimizer in the target domain. Assume that $\|x\|_2 \leq R, \forall x \in \mathcal{X}$, and that the loss function obeys the triangle inequality and is $\rho$-Lipschitz and convex. Then, for any $\delta \in (0, 1)$, with probability at least $1 - \delta$, we have:*

$$\mathcal{L}_{\mathcal{D}_\mathcal{T}}(h_S) \leq \mathcal{L}_{\mathcal{D}_\mathcal{T}}(h^*) + \varepsilon'_\Gamma + 2\|\Gamma^\mathcal{S}\|_1 \operatorname{dist}_{\mathcal{Y}}(\mathcal{D}_\mathcal{T}, \mathcal{D}_\mathcal{S}) \ , \tag{3}$$

*where*

$$\varepsilon'_\Gamma = \min \left\{ \frac{\|\Gamma\|_\infty \rho^2 R^2}{\lambda} + \left( \frac{\rho^2 R^2 (\|\Gamma\|_2^2 + \|\Gamma\|_\infty)}{\lambda} + \left( \frac{\|\Gamma\|_2}{\sqrt{N}} + \|\Gamma\|_\infty \right) B(\Gamma) \right) \sqrt{\frac{N \log \frac{4}{\delta}}{2}}, \right.$$

$$\left. \frac{2\|\Gamma\|_\infty \|\Gamma\|_2 \rho^2 R^2}{\lambda} \sqrt{2N \log \frac{8}{\delta}} + 2\|\Gamma\|_2 B(\Gamma) \sqrt{\frac{\log \frac{4}{\delta}}{2}} \right\}.$$

**Lemma A.** *Let $\mathcal{D}$ be a distribution over $\mathcal{X} \times \mathcal{Y}$ and let $\hat{\mathcal{D}}$ be the corresponding empirical distribution for a sample $S = \{(x_i, y_i)\}_{i=1}^N$. Then, for any $\delta > 0$, with probability at least $1 - \delta$, the following hold:*

$$\operatorname{dist}_{\mathcal{Y}}(\mathcal{D}_\mathcal{S}, \mathcal{D}_\mathcal{T}) \leq \operatorname{dist}_{\mathcal{Y}}(\hat{\mathcal{D}}_\mathcal{S}, \hat{\mathcal{D}}_\mathcal{T}) + \frac{4\rho^2 R^2}{\lambda} \left( \frac{1}{N_\mathcal{S}} + \frac{1}{N_\mathcal{T}} \right) \sqrt{2 \log \frac{4}{\delta}} + B \left( \sqrt{\frac{\log \frac{4}{\delta}}{2N_\mathcal{S}}} + \sqrt{\frac{\log \frac{4}{\delta}}{2N_\mathcal{T}}} \right)$$

*for $\rho$-Lipschitz continuous loss, and*

$$\mathrm{dist}_{\mathcal{Y}}(\mathcal{D}_{\mathcal{S}}, \mathcal{D}_{\mathcal{T}}) \leq \mathrm{dist}_{\mathcal{Y}}(\hat{\mathcal{D}}_{\mathcal{S}}, \hat{\mathcal{D}}_{\mathcal{T}}) + \frac{4R}{\lambda}\left(\frac{1}{N_{\mathcal{S}}} + \frac{1}{N_{\mathcal{T}}}\right)\sqrt{2\log\frac{4}{\delta}} + \left(\sqrt{\frac{\log\frac{4}{\delta}}{2N_{\mathcal{S}}}} + \sqrt{\frac{\log\frac{4}{\delta}}{2N_{\mathcal{T}}}}\right)$$

*for 0-1 loss.*

**Remark 4.** *Lemma A shows that the discrepancy distance between two distributions $\mathcal{D}_{\mathcal{S}}$ and $\mathcal{D}_{\mathcal{T}}$ can be estimated from finite samples $\hat{\mathcal{D}}_{\mathcal{S}}$ and $\hat{\mathcal{D}}_{\mathcal{T}}$. However, we still have to find an approach to compute the empirical $\mathcal{Y}$-discrepancy $\mathrm{dist}_{\mathcal{Y}}(\hat{\mathcal{D}}_{\mathcal{S}}, \hat{\mathcal{D}}_{\mathcal{T}})$, which is given by Lemma B*

**Lemma B.** *Let $S_{\mathcal{S}} = \{x_i^{\mathcal{S}}, y_i^{\mathcal{S}}\}_{i=1}^{N_{\mathcal{S}}}$ and $S_{\mathcal{T}} = \{x_i^{\mathcal{T}}, y_i^{\mathcal{T}}\}_{i=1}^{N_{\mathcal{T}}}$, respectively, be the training samples of source and target domains, $\hat{\mathcal{D}}_{\mathcal{S}}$ and $\hat{\mathcal{D}}_{\mathcal{T}}$ be their corresponding empirical distributions. Then, we have*

$$\mathrm{dist}_{\mathcal{Y}}(\hat{\mathcal{D}}_{\mathcal{S}}, \hat{\mathcal{D}}_{\mathcal{T}}) = 1 - \min_{h \in \mathcal{H}}\left[\frac{1}{N_{\mathcal{T}}}\sum_{x_i^{\mathcal{T}}:y_i^{\mathcal{T}}=1}\mathbb{1}_{h(x_i^{\mathcal{T}})=1} + \frac{1}{N_{\mathcal{S}}}\sum_{x_i^{\mathcal{S}}:y_i^{\mathcal{S}}=0}\mathbb{1}_{h(x_i^{\mathcal{S}})=1}\right.$$
$$\left. + \frac{1}{N_{\mathcal{T}}}\sum_{x_i^{\mathcal{T}}:y_i^{\mathcal{T}}=0}\mathbb{1}_{h(x_i^{\mathcal{T}})=0} + \frac{1}{N_{\mathcal{S}}}\sum_{x_i^{\mathcal{S}}:y_i^{\mathcal{S}}=1}\mathbb{1}_{h(x_i^{\mathcal{S}})=0}\right]$$

**Remark 5.** *Lemma B shows that, for 0-1 loss, $\mathrm{dist}_{\mathcal{Y}}(\hat{\mathcal{D}}_{\mathcal{S}}, \hat{\mathcal{D}}_{\mathcal{T}})$ can be computed by constructing a new classification problem, where the positive target examples and negative source examples are positively labeled, and the negative target examples and positive source examples are negatively labeled. Then, $\mathrm{dist}_{\mathcal{Y}}(\hat{\mathcal{D}}_{\mathcal{S}}, \hat{\mathcal{D}}_{\mathcal{T}})$ can by computed by finding the hypothesis that minimizes 0-1 loss of the new classification problem.*

**Proposition A.** *Let $f(x) = \sum_{k=1}^{K}\alpha_k h_k(x)$ be the ensemble of classifiers returned by* `gapBoost`*, with each base learner trained by solving (1). For simplicity, we assume that $\sum_{k=1}^{K}\alpha_k = 1$. Then, for any $\delta \in (0, 1)$, with probability at least $1 - \delta$, we have*

$$\mathcal{L}_{\mathcal{D}_{\mathcal{T}}}(f) \leq \mathcal{L}_{S_{\mathcal{T}}}(f) + \frac{2\rho^2 R^2 \gamma_{\infty}^{\mathcal{T}}}{\lambda}\sqrt{2\log\frac{4}{\delta}} + B(\Gamma)\sqrt{\frac{\log\frac{2}{\delta}}{2N_{\mathcal{T}}}} \ .$$

*where $\gamma_{\infty}^{\mathcal{T}}$ is the largest weight of the target sample over all boosting iterations.*

## 3  Proof of the Results

### 3.1  Step 1: Bound $\mathcal{L}_{\mathcal{D}_{\mathcal{T}}}$ from $\mathcal{L}_{\mathcal{D}}^{\Gamma}$

**Lemma 3.** *Let $\mathcal{L}_{\mathcal{D}}^{\Gamma} = \mathcal{L}_{\mathcal{D}_{\mathcal{T}}}^{\Gamma^{\mathcal{T}}} + \mathcal{L}_{\mathcal{D}_{\mathcal{S}}}^{\Gamma^{\mathcal{S}}}$ be the expected weighted loss of $\mathcal{L}_{S}^{\Gamma}$, and let $h^*$ be the optimal hypothesis that minimizes the error $h^* = \arg\min_{h \in \mathcal{H}}\mathcal{L}_{\mathcal{D}_{\mathcal{T}}}(h) + \mathcal{L}_{\mathcal{D}_{\mathcal{S}}}(h)$. Then, for any $h \in \mathcal{H}$, we have*

$$\mathcal{L}_{\mathcal{D}_{\mathcal{T}}}(h) \leq \mathcal{L}_{\mathcal{D}}^{\Gamma} + \|\Gamma^{\mathcal{S}}\|_1 \mathrm{dist}_{\mathcal{Y}}(\mathcal{D}_{\mathcal{T}}, \mathcal{D}_{\mathcal{S}}).$$

*Proof.*

$$\mathcal{L}_{\mathcal{D}_{\mathcal{T}}}(h) - \mathcal{L}_{\mathcal{D}}^{\Gamma}(h)$$
$$\leq \left|\mathcal{L}_{\mathcal{D}_{\mathcal{T}}}(h) - \mathcal{L}_{\mathcal{D}_{\mathcal{T}}}^{\Gamma^{\mathcal{T}}}(h) - \mathcal{L}_{\mathcal{D}_{\mathcal{S}}}^{\Gamma^{\mathcal{S}}}(h)\right|$$
$$= |\mathcal{L}_{\mathcal{D}_{\mathcal{T}}}(h) - (1 - \gamma_{\mathcal{S}})\mathcal{L}_{\mathcal{D}_{\mathcal{T}}}(h) - \gamma_{\mathcal{S}}\mathcal{L}_{\mathcal{D}_{\mathcal{S}}}(h)| \qquad \text{linearity of expectation}$$
$$= \|\Gamma^{\mathcal{S}}\|_1 |\mathcal{L}_{\mathcal{D}_{\mathcal{T}}}(h) - \mathcal{L}_{\mathcal{D}_{\mathcal{S}}}(h)|$$
$$\leq \|\Gamma^{\mathcal{S}}\|_1 \mathrm{dist}_{\mathcal{Y}}(\mathcal{D}_{\mathcal{T}}, \mathcal{D}_{\mathcal{S}}) \qquad \text{definition of } \mathcal{Y}\text{-discrepancy distance}$$

$$\square$$

## 3.2 Step 2: Bound $\mathcal{L}_{\mathcal{D}}^{\Gamma}$ from $\mathcal{L}_{S}^{\Gamma}$

To bound $\mathcal{L}_{\mathcal{D}}^{\Gamma}(h)$, we need some auxiliary results. In Section 3.2.1, we bound the generalization performance within the framework of algorithmic stability. In particular, we show that the generalization loss of *weight dependent uniformly stable* algorithms can be bounded. In Section 3.2.2, we develop another bound by leveraging the connection between *weighted Rademacher complexity* and *augment algorithmic stability* [7].

### 3.2.1 Algorithmic Stability Bound

**Definition A** (**Weight dependent uniform stability**). *Let $h_S \in \mathcal{H}$ be the hypothesis returned by a learning algorithm $\mathcal{A}$ when trained on sample $S$ weighted by $\Gamma$. An algorithm $\mathcal{A}$ has weight dependent uniform stability, with $\beta_i \geq 0$, if the following holds:*

$$\sup_{z \sim \mathcal{D}} |\ell(h_S(x), y) - \ell(h_{S^i}(x), y)| \leq \beta_i, \qquad \forall S, S^i$$

*where $S^i$ is the training sample $S$ with the $i$-th example $z_i$ replaced by an i.i.d. example $z_i'$.*

**Remark 6.** *By letting $\beta = \max\{\beta_i\}_{i=1}^{N}$, it is trivial to show that weight dependent uniform stability implies uniform stability.*

Next, we bound the generalization error for weight dependent stable algorithms.

**Lemma 4.** *Assume that the loss function is upper bounded by $B \geq 0$. Let $S$ be a training sample of $N$ i.i.d. points drawn from some distribution $\mathcal{D}$, weighted by $\Gamma$, and let $h_S$ be the hypothesis returned by a weight dependent stable learning algorithm $\mathcal{A}$. Then, for any $\delta \in (0, 1)$, with probability at least $1 - \delta$, the following holds:*

$$\mathcal{L}_{\mathcal{D}}^{\Gamma}(h_S) \leq \mathcal{L}_{S}^{\Gamma}(h_S) + \beta + (\Delta + \beta + \|\Gamma\|_{\infty} B) \sqrt{\frac{N \log \frac{1}{\delta}}{2}},$$

*where $\Delta = \sum_{i=1}^{N} \gamma_i \beta_i$ and $\|\Gamma\|_{\infty} = \max\{\gamma_i\}_{i=1}^{N}$.*

*Proof.* Let $\Phi^{\Gamma}(S) = \mathcal{L}_{\mathcal{D}}^{\Gamma}(h_S) - \mathcal{L}_{S}^{\Gamma}(h_S)$. Then, by the definition of $\Phi^{\Gamma}$, we have

$$|\Phi(S) - \Phi(S^i)| \leq |\mathcal{L}_{\mathcal{D}}^{\Gamma}(h_S) - \mathcal{L}_{\mathcal{D}}^{\Gamma}(h_{S^i})| + |\mathcal{L}_{S}^{\Gamma}(h_S) + \mathcal{L}_{S^i}^{\Gamma}(h_{S^i})|$$

By the stability of the algorithm, we have[3]

$$|\mathcal{L}_{\mathcal{D}}^{\Gamma}(h_S) - \mathcal{L}_{\mathcal{D}}^{\Gamma}(h_{S^i})| = |\mathbb{E}_{z \sim \mathcal{D}}[\ell_z(h_S)] - \mathbb{E}_{z \sim \mathcal{D}}[\ell_z(h_{S^i})]| \leq \beta,$$

where $\beta = \max\{\beta_i\}_{i=1}^{N}$. In addition, we also have

$$
\begin{aligned}
|\mathcal{L}_{S}^{\Gamma}(h_S) - \mathcal{L}_{S^i}^{\Gamma}(h_{S^i})| &= \left| \sum_{j \neq i} \gamma_j (\ell_{z_j}(h_S) - \ell_{z_j}(h_{S^i})) + \gamma_i (\ell_{z_i}(h_S) - \ell_{z_i'}(h_{S^i})) \right| \\
&\leq \left| \sum_{j \neq i} \gamma_j \left| \ell_{z_j}(h_S) - \ell_{z_j}(h_{S^i}) \right| + \gamma_i \left| \ell_{z_i}(h_S) - \ell_{z_i'}(h_{S^i}) \right| \right| \\
&\leq \sum_{j \neq i} \gamma_j \beta_j + \gamma_i B \\
&\leq \Delta + \|\Gamma\|_{\infty} B
\end{aligned}
$$

Consequently, $\Phi^{\Gamma}$ satisfies $|\Phi^{\Gamma}(S) - \Phi^{\Gamma}(S^i)| \leq \sum_{i=1}^{N} \gamma_i \beta_i + \beta + \|\Gamma\|_{\infty} B$. By applying McDiarmid's inequality, we have

$$\Pr[\Phi(S) \geq \epsilon + \mathbb{E}[\Phi(S)]] \leq \exp\left( \frac{-2\epsilon^2}{N (\Delta + \beta + \|\Gamma\|_{\infty} B)^2} \right). \tag{4}$$

By setting $\delta = \exp\left(\frac{-2\epsilon^2}{N(\Delta+\beta+\|\Gamma\|_\infty B)^2}\right)$, we obtain $\epsilon = (\Delta + \beta + \|\Gamma\|_\infty B)\sqrt{\frac{N\log\frac{1}{\delta}}{2}}$. Plugging $\epsilon$ back to (4) and rearranging terms, with probability $1 - \delta$, we have

$$\Phi(S) \leq \mathbb{E}[\Phi(S)] + (\Delta + \beta + \|\Gamma\|_\infty B)\sqrt{\frac{N\log\frac{1}{\delta}}{2}}. \tag{5}$$

By the linearity of expectation, we have $\mathbb{E}[\Phi(S)] = \mathbb{E}_{S\sim\mathcal{D}^N}[\mathcal{L}_\mathcal{D}^\Gamma(h_S)] - \mathbb{E}_{S\sim\mathcal{D}^N}[\mathcal{L}_S^\Gamma(h_S)]$. By the definition of the generalization error, we have

$$\mathbb{E}_{S\sim\mathcal{D}^N}[\mathcal{L}_\mathcal{D}^\Gamma(h_S)] = \mathbb{E}_{S,z\sim\mathcal{D}^{N+1}}[\ell_z(h_S)].$$

On the other hand, by the linearity of expectation, we have

$$\mathbb{E}_{S\sim\mathcal{D}^N}[\mathcal{L}_S^\Gamma(h_S)] = \mathbb{E}_{S\sim\mathcal{D}^N}\left[\sum_{i=1}^N \gamma_i \ell_{z_i}(h_S)\right] = \mathbb{E}_{S,z\sim\mathcal{D}^{N+1}}[\ell_z(h_{S'})],$$

where $S'$ is a sample of $N$ data points containing $z$ drawn from the data set $\{S, z\}$. Therefore, we have

$$\begin{aligned}
\mathbb{E}[\Phi(S)] &\leq \left|\mathbb{E}_{S,z\sim\mathcal{D}^{N+1}}[\ell_z(h_S)] - \mathbb{E}_{S,z\sim\mathcal{D}^{N+1}}[\ell_z(h_{S'})]\right| \\
&\leq \mathbb{E}_{S,z\sim\mathcal{D}^{N+1}}[|\ell_z(h_S) - \ell_z(h_{S'})|] && \text{Jensen's inequality} \\
&\leq \beta.
\end{aligned}$$

Replacing $\mathbb{E}[\Phi(S)]$ by $\beta$ in Eq. 5 completes the proof. $\qquad\square$

Lemma 5 shows that the algorithm solving (1) has weight dependent stability.

**Lemma 5.** *The learning algorithm (1) with a $\rho$-Lipschitz continuous loss function and the regularizer $\mathcal{R}(w) = \|w\|_2^2$ has weight dependent uniform stability, with*

$$\beta_i \leq \frac{\gamma_i \rho^2 R^2}{\lambda}.$$

*Proof.* Let $\mathcal{V}_S(w) = \mathcal{L}_S^\Gamma(w) + \lambda\mathcal{R}(w)$. By the definition of Bregman divergence, we have

$$\begin{aligned}
d_{\mathcal{V}_{S^i}}(w_S, w_{S^i}) + d_{\mathcal{V}_S}(w_{S^i}, w_S) &= \mathcal{L}_{S^i}^\Gamma(w_S) - \mathcal{L}_{S^i}^\Gamma(w_{S^i}) + \mathcal{L}_S^\Gamma(w_{S^i}) - \mathcal{L}_S^\Gamma(w_S) \\
&= \gamma_i\left(\ell(\langle w_S, x_i'\rangle, y_i') - \ell(\langle w_{S^i}, x_i'\rangle, y_i') + \ell(\langle w_{S^i}, x_i\rangle, y_i) - \ell(\langle w_S, x_i\rangle, y_i)\right) \\
&\leq \gamma_i\left(\rho\,|\langle w_S - w_{S^i}, x_i'\rangle| + |\rho\langle w_S - w_{S^i}, x_i\rangle|\right) \\
&\leq 2\gamma_i\rho R\,\|w_S - w_{S^i}\|_2
\end{aligned}$$

where $w_S$ and $w_{S^i}$ are, respectively, the optimal solutions of $\mathcal{V}_S$ and $\mathcal{V}_{S^i}$. The first equality holds because of the first-order optimality condition [3] of $\mathcal{V}_S$ and $\mathcal{V}_{S^i}$, and the last two inequalities are, respectively, due to the Lipschitz continuity of loss function $\ell$ and the Cauchy-Schwarz inequality.

Since $d_{\lambda\mathcal{R}}(w_S, w_{S^i}) = d_{\lambda\mathcal{R}}(w_{S^i}, w_S) = \lambda\|w_S - w_{S^i}\|_2^2$, by the non-negative and additive properties of Bregman divergence, we have

$$\lambda\|w_S - w_{S^i}\|_2^2 \leq \gamma_i\rho R\,\|w_S - w_{S^i}\|_2,$$

which gives

$$\|w_S - w_{S^i}\|_2 \leq \frac{\gamma_i\rho R}{\lambda}.$$

Consequently, by the Lipschitz continuity of $\ell$ and the Cauchy-Schwarz inequality, we have

$$\beta_i \leq \frac{\gamma_i\rho^2 R^2}{\lambda}.$$

$\qquad\square$

Combining Lemma 4 and Lemma 5, we immediately obtain obtained the generalization bound of the learning algorithms solving (1).

**Theorem 1.** *Assume that the loss function is $\rho$-Lipschitz continuous and upper bounded by $B \geq 0$. Let $S$ be a training sample of $N$ i.i.d. points drawn from some distribution $\mathcal{D}$, weighted by $\Gamma$, and let $w_S$ be the hypothesis returned by a learning algorithm $\mathcal{A}$ that minimizes (1) with the regularizer $\mathcal{R}(w) = \|w\|_2^2$. Then, for any $\delta \in (0,1)$, with probability at least $1 - \delta$, the following holds:*

$$\mathcal{L}_{\mathcal{D}}^{\Gamma}(w_S) \leq \mathcal{L}_S^{\Gamma}(w_S) + \frac{\|\Gamma\|_{\infty}\rho^2 R^2}{\lambda} + \left( \frac{\rho^2 R^2(\|\Gamma\|_2^2 + \|\Gamma\|_{\infty})}{\lambda} + \|\Gamma\|_{\infty}B \right) \sqrt{\frac{N \log \frac{1}{\delta}}{2}}, \quad (6)$$

**Remark 7.** *If $\gamma_i = \frac{1}{N}, \forall i = \{1, \ldots, N\}$, we recover the standard stability bound from (6) [13]. In addition, $\|\Gamma\|_{\infty}$ and $\|\Gamma\|_2^2$ in (6) also imply that one should assign equal weights to all the instances. In Section 3.3, we will show that in the setting of instance weighting for transfer learning, the model complexity (hence the loss function) can also be upper bounded as a function of $\Gamma$, and therefore, $\gamma_i = \frac{1}{N}$ may not be an optimal weighting scheme for transfer learning.*

### 3.2.2   Rademacher Complexity Bound

In this section, we develop the Rademacher complexity bound for the learning algorithm (1).

**Definition B (Weighted Rademacher complexity).** *Let $\mathcal{H}$ be a hypothesis class defined over a set $\mathcal{X}$ drawn from a distribution $\mathcal{D}$, $S = \{x_i\}_{i=1}^N$ a fixed sample of size $N$ with elements in $\mathcal{X}$, and $\Gamma = \{\gamma_i\}_{i=1}^N$ be the weights for sample. Then, the weighted Rademacher complexity of $\mathcal{H}$ is defined as*

$$\mathfrak{R}_{\mathcal{D}}^{\Gamma}(\mathcal{H}) = \mathbb{E} \sup_{h \in \mathcal{H}} \sum_{i=1}^N \sigma_i \gamma_i h(x_i),$$

*where $\sigma_i, \ldots, \sigma_N$ are are independent uniform random variables taking values in $\{-1, +1\}$.*

**Lemma 6.** *Assume that the loss function is $\rho$-Lipschitz continuous and upper bounded by $B \geq 0$. Let $S$ be a training sample of $N$ i.i.d. points drawn from some distribution $\mathcal{D}$, weighted by $\Gamma$. Then, for any $\delta \in (0,1)$, with probability at least $1 - \delta$, the following holds for any $h \in \mathcal{H}$:*

$$\mathcal{L}_{\mathcal{D}}^{\Gamma}(h) \leq \mathcal{L}_S^{\Gamma}(h) + 2\rho \mathfrak{R}_{\mathcal{D}}^{\Gamma}(\mathcal{H}) + B\|\Gamma\|_2 \sqrt{\frac{\log \frac{1}{\delta}}{2}}. \quad (7)$$

*Proof.* For each hypothesis $h \in \mathcal{H}$, let $\mathcal{G}$ be a family of functions mapping $z \in \mathcal{X} \times \mathcal{Y}$ to some loss function $\ell_z(h)$:

$$\mathcal{G} = \mathcal{H} \circ \ell = \{z \in \mathcal{X} \times \mathcal{Y} \to \ell_z(h) : h \in \mathcal{H}\}.$$

Let $\Phi(S) = \sup_{g \in \mathcal{G}} \mathbb{E}^{\Gamma}[g] - \hat{\mathbb{E}}_S^{\Gamma}[g]$, where $\hat{\mathbb{E}}_S^{\Gamma}(g) = \sum_{i=1}^N \gamma_i g(z_i)$, and $\mathbb{E}^{\Gamma}[g] = \mathbb{E}_S\left[\mathbb{E}^{\Gamma}[g]\right]$. Since the loss function is bounded by $B$, we have $|\Phi(S) - \Phi(S^i)| \leq \gamma_i B$. Then, by applying McDiarmid's inequality, for any $\delta \geq 0$, with probability at least $1 - \delta$, the following holds:

$$\Phi(S) \leq \mathbb{E}[\Phi(S)] + B\sqrt{\frac{\log \frac{1}{\delta} \sum_{i=1}^N \gamma_i^2}{2}}$$

Next, by Talagrand's Lemma [6] and the similar proof scheme of Theorem 3.1 in [13], we can show that for a $\rho$-Lipschitz loss function, we have

$$\mathbb{E}[\Phi(S)] = \mathbb{E}_S \left[ \sup_{g \in \mathcal{G}} \mathbb{E}^{\Gamma}[g] - \hat{\mathbb{E}}_S^{\Gamma}[g] \right]$$

$$= \mathbb{E}_S \left[ \sup_{g \in \mathcal{G}} \mathbb{E}_{\tilde{S}} \left[ \hat{\mathbb{E}}_{\tilde{S}}^{\Gamma}[g] - \hat{\mathbb{E}}_S^{\Gamma}[g] \right] \right]$$

$$\leq \mathbb{E}_{S,\tilde{S}} \left[ \sup_{g \in \mathcal{G}} \hat{\mathbb{E}}_{\tilde{S}}^{\Gamma}[g] - \hat{\mathbb{E}}_S^{\Gamma}[g] \right]$$

$$= \mathbb{E}_{S,\tilde{S}} \left[ \sup_{g \in \mathcal{G}} \sum_{i=1}^{N} \gamma_i \left( g(\tilde{z}_i) - g(z_i) \right) \right]$$

$$= \mathbb{E}_{S,\tilde{S},\sigma} \left[ \sup_{g \in \mathcal{G}} \sum_{i=1}^{N} \sigma_i \gamma_i \left( g(\tilde{z}_i) - g(z_i) \right) \right]$$

$$\leq \mathbb{E}_{\tilde{S},\sigma} \left[ \sup_{g \in \mathcal{G}} \sum_{i=1}^{N} \sigma_i \gamma_i g(\tilde{z}_i) \right] + \mathbb{E}_{S,\sigma} \left[ \sup_{g \in \mathcal{G}} \sum_{i=1}^{N} -\sigma_i \gamma_i g(z_i) \right]$$

$$= 2 \mathbb{E}_{S,\sigma} \left[ \sup_{g \in \mathcal{G}} \sum_{i=1}^{N} \sigma_i \gamma_i g(z_i) \right] = 2 \mathfrak{R}_{\mathcal{D}}^{\Gamma}(\mathcal{G}) \leq 2\rho \mathfrak{R}_{\mathcal{D}}^{\Gamma}(\mathcal{H})$$

Therefore, with probability at least $1 - \delta$, the following holds

$$\mathcal{L}_{\mathcal{D}}^{\Gamma}(h) \leq \mathcal{L}_{S}^{\Gamma}(h) + 2\rho \mathfrak{R}_{\mathcal{D}}^{\Gamma}(\mathcal{H}) + B\|\Gamma\|_2 \sqrt{\frac{\log \frac{1}{\delta}}{2}}.$$

$\square$

From the proof of Lemma 5, we can show that the algorithm solving (1) has uniform augment stability.

**Corollary 1.** *The learning algorithm (1) with a $\rho$-Lipschitz continuous loss function and the regularizer $R(w) = \|w\|_2^2$ has uniform argument stability, with*

$$\alpha \leq \frac{\|\Gamma\|_{\infty} \rho R}{\lambda}$$

Next, we show that the weighted Rademacher complexity $\mathfrak{R}_{\mathcal{D}}^{\Gamma}(\mathcal{B})$ of an algorithmic hypothesis class can be upper bounded.

**Lemma 7.** *Let $S$ be a training sample of $N$ i.i.d. points drawn from some distribution $\mathcal{D}$, weighted by $\Gamma$, and let $h_S$ be the linear hypothesis that has $\alpha$-uniform argument stability. Then, for any $\delta \in (0,1)$, with probability at least $1 - \delta$, the weighted Rademacher complexity $\mathfrak{R}_{\mathcal{D}}^{\Gamma}(\mathcal{B})$ can be upper bounded by*

$$\mathfrak{R}_{\mathcal{D}}^{\Gamma}(\mathcal{B}) \leq \alpha R \|\Gamma\|_2 \sqrt{2N \log \frac{2}{\delta}}. \tag{8}$$

*Proof.* We follow the similar proof scheme of Theorem 1 in [7]

$$\mathfrak{R}_{\mathcal{D}}^{\Gamma}(\mathcal{B}) = \mathbb{E} \sup_{h \in \mathcal{B}} \sum_{i=1}^{N} \sigma_i \gamma_i \langle h, x_i \rangle$$

$$= \mathbb{E} \sup_{h \in \mathcal{B}} \sum_{i=1}^{N} \left( \sigma_i \gamma_i \langle h, x_i \rangle - \sigma_i \gamma_i \langle \mathbb{E} h_S, x_i \rangle \right) \qquad \mathbb{E} h_S \text{ is a constant}$$

$$= \mathbb{E} \sup_{h \in \mathcal{B}} \sum_{i=1}^{N} \sigma_i \gamma_i \langle h - \mathbb{E} h_S, x_i \rangle$$

$$\leq \mathbb{E} \sup_{h \in \mathcal{B}} \| h - \mathbb{E} h_S \| \left\| \sum_{i=1}^{N} \sigma_i \gamma_i x_i \right\| \qquad \text{Cauchy-Schwarz inequality}$$

$$\leq \alpha \sqrt{2N \log \frac{2}{\delta}} \left( \sum_{i=1}^{N} \| \gamma_i x_i \|_2^2 \right)^{\frac{1}{2}} \qquad \text{Lemma 1}$$

$$\leq \alpha \sqrt{2N \log \frac{2}{\delta}} \left( R^2 \sum_{i=1}^{N} \gamma_i^2 \right)^{\frac{1}{2}}$$

$$= \alpha R \| \Gamma \|_2 \sqrt{2N \log \frac{2}{\delta}}$$

$\square$

Combining Lemma 6, Corollary 1, and Lemma 7, we obtain another generalization bound of the learning algorithm solving (1).

**Theorem 2.** *Assume that the loss function is $\rho$-Lipschitz continuous and upper bounded by $B \geq 0$. Let $S$ be a training sample of $N$ i.i.d. points drawn from some distribution $\mathcal{D}$, weighted by $\Gamma$, and let $w_S$ be the hypothesis returned by a $\beta$-stable learning algorithm $\mathcal{A}$. Then, for any $\delta \in (0, 1)$, with probability at least $1 - \delta$, we have*

$$\mathcal{L}_{\mathcal{D}}^{\Gamma}(w_S) \leq \mathcal{L}_{S}^{\Gamma}(w_S) + \frac{2 \| \Gamma \|_{\infty} \| \Gamma \|_2 \rho^2 R^2}{\lambda} \sqrt{2N \log \frac{4}{\delta}} + B \| \Gamma \|_2 \sqrt{\frac{\log \frac{2}{\delta}}{2}}. \qquad (9)$$

**Remark 8.** *If $\gamma_i = \frac{1}{N}, \forall i = \{1, \ldots, N\}$, we recover the standard argument stability bound from (9) [7]. Similar to Theorem 1, Theorem 2 also suggests assigning equal weights to the instances. If $\| \Gamma \|_{\infty}$ (hence $\| \Gamma \|_2^2$) is of order $\mathcal{O}(\frac{1}{n})$, both Theorem 1 and Theorem 2 lead to a convergence rate of the generalization bound is of order $\mathcal{O}(\frac{1}{\sqrt{n}})$. In the setting of transfer learning, it is usually the case that $N_{\mathcal{T}} \ll N_{\mathcal{S}}$. Consequently, we may have $\| \Gamma \|_{\infty} \ll \frac{1}{N_{\mathcal{T}}}$, which implies the benefits of transfer learning compared to single task learning.*

By combining Lemma 3 with Theorems 1 – 2, we immediately obtain Theorem A.

### 3.3 Step 3: Bound $B$

**Definition C (Performance gap).** *Let $\mathcal{V}_{\mathcal{S}}(h) = \mathcal{L}_{S_{\mathcal{S}}}^{\Gamma^{\mathcal{S}}}(h) + \eta \lambda \mathcal{R}(h)$ and $\mathcal{V}_{\mathcal{T}}(h) = \mathcal{L}_{S_{\mathcal{T}}}^{\Gamma^{\mathcal{T}}}(h) + \eta \lambda \mathcal{R}(h)$, respectively, be the objective functions in the source and target domains, where $\eta \in (0, \frac{1}{2})$, and let their minimizers, respectively, be $h_{S_{\mathcal{S}}}$ and $h_{S_{\mathcal{T}}}$. The performance gap between the source and target domains is defined as*

$$\nabla = \nabla_{\mathcal{T}} + \nabla_{\mathcal{S}},$$

*where $\nabla_{\mathcal{S}} = \mathcal{L}_{S_{\mathcal{S}}}^{\Gamma^{\mathcal{S}}}(h_{S_{\mathcal{T}}}) - \mathcal{L}_{S_{\mathcal{S}}}^{\Gamma^{\mathcal{S}}}(h_{S_{\mathcal{S}}})$ and $\nabla_{\mathcal{T}} = \mathcal{L}_{S_{\mathcal{T}}}^{\Gamma^{\mathcal{T}}}(h_{S_{\mathcal{S}}}) - \mathcal{L}_{S_{\mathcal{T}}}^{\Gamma^{\mathcal{T}}}(h_{S_{\mathcal{T}}})$.*

The following lemma shows that the model complexity of the transfer learning algorithm (1) can be upper bounded in terms of the performance $\nabla$.

**Lemma C.** *Let $h_S$ be the optimal solution of the instance weighting transfer learning problem (1). Then, we have:*

$$\|h_S\|_2 \leq \sqrt{\frac{\nabla}{2\lambda(1-2\eta)} + \frac{\|h_{S_S}\|_2^2 + \|h_{S_T}\|_2^2}{2}} \ .$$

*Proof.* By the definition of $w_{S_T}$, $w_{S_S}$, and $w_S$, we have

$$\mathcal{V}_S(w_{S_S}) \leq \mathcal{V}_S(w_S), \quad \text{and} \quad \mathcal{V}_T(w_{S_T}) \leq \mathcal{V}_T(w_S),$$

which gives

$$\mathcal{V}_S(w_{S_S}) + \mathcal{V}_T(w_{S_T}) \leq \mathcal{V}(w_S) + (2\eta - 1)\lambda\mathcal{R}(w_S). \tag{10}$$

On the other hand, we also have

$$\mathcal{V}(w_S) \leq \mathcal{V}(w_{S_S}), \tag{11}$$

and

$$\mathcal{V}(w_S) \leq \mathcal{V}(w_{S_T}). \tag{12}$$

From (10) and (11), we have

$$\begin{aligned}
\lambda(1-2\eta)\mathcal{R}(w_S) &\leq \mathcal{V}(w_S) - \mathcal{V}_S(w_{S_S}) - \mathcal{V}_T(w_{S_T}) \tag{13}\\
&\leq \mathcal{V}(w_{S_S}) - \mathcal{V}_S(w_{S_S}) - \mathcal{V}_T(w_{S_T})\\
&= \nabla_T + \lambda(1-\eta)\mathcal{R}(w_{S_S}) - \eta\lambda\mathcal{R}(w_{S_T}).
\end{aligned}$$

Similarly, from (10) and (12), we also have

$$\lambda(1-2\eta)\mathcal{R}(w_S) \leq \nabla_S + \lambda(1-\eta)\mathcal{R}(w_{S_T}) - \eta\lambda\mathcal{R}(w_{S_S}) \tag{14}$$

Combining (13) and (14) gives

$$\mathcal{R}(w_S) \leq \frac{\nabla}{2\lambda(1-2\eta)} + \frac{1}{2}\left(\mathcal{R}(w_{S_S}) + \mathcal{R}(w_{S_T})\right).$$

Substituting $\mathcal{R}(w) = \|w\|_2^2$ concludes the proof. $\square$

Substituting the hinge loss for classification into the loss function, we immediately obtain the following corollary.

**Corollary 2.** *The hinge loss function of the transfer learning algorithm (1) for classification with the regularizer $\mathcal{R}(w) = \|w\|_2^2$ can be upper bounded by*

$$B \leq 1 + R\sqrt{\frac{\nabla}{2\lambda(1-2\eta)} + \frac{1}{2}\left(\|w_S\|_2^2 + \|w_T\|_2^2\right)}.$$

Similarly, Corollary 2 bounds the $\ell_q$ loss for robust regression.

**Corollary 3.** *Assume that $|y| \leq Y, \forall y \in \mathcal{Y}$. Then, the $\ell_q$ loss function of the transfer learning algorithm (1) for regression with the regularizer $\mathcal{R}(w) = \|w\|_2^2$ can be upper bounded by*

$$B \leq \left(Y + R\sqrt{\frac{\nabla}{2\lambda(1-2\eta)} + \frac{1}{2}\left(\|w_S\|_2^2 + \|w_T\|_2^2\right)}\right)^q.$$

### 3.4 Proof of Corollary A

**Lemma 8** (Weighted Hoeffding's inequality). *Let $\{X_i\}_{i=1}^N$ be independent random variables with $a_i \leq X_i \leq b_i$. Then, for any $\epsilon \geq 0$, the following inequality holds for $S = \sum_{i=1}^N \gamma_i X_i$:*

$$Pr\left[|S - \mathbb{E}[S]| \geq \epsilon\right] \leq 2e^{-2\epsilon^2 / \sum_{i=1}^N \gamma_i^2(b_i - a_i)^2}.$$

*Proof.*

$$\Pr\left[S - \mathbb{E}[S] \geq \epsilon\right] \leq e^{t\epsilon}\mathbb{E}\left[e^{t(S-\mathbb{E}[S])}\right]$$

$$= \prod_{i=1}^{N} e^{t\epsilon}\mathbb{E}\left[e^{t\gamma_i(X_i-\mathbb{E}[X_i])}\right]$$

$$\leq \prod_{i=1}^{N} e^{t\epsilon}e^{t^2\gamma_i^2(b_i-a_i)^2/8}$$

$$= e^{t\epsilon}e^{t^2}e^{\sum_{i=1}^{N}\gamma_i^2(b_i-a_i)^2/8}$$

$$\leq e^{-2\epsilon^2/\sum_{i=1}^{N}\gamma_i^2(b_i-a_i)^2},$$

where we have chosen $t = \frac{4\epsilon}{\sum_{i=1}^{N}\gamma_i^2(b_i-a_i)^2}$ as the minimizer. Similarly, we can also show that

$$\Pr\left[S - \mathbb{E}[S] \leq -\epsilon\right] \leq e^{-2\epsilon^2/\sum_{i=1}^{N}\gamma_i^2(b_i-a_i)^2},$$

$\square$

Given Lemma 8, we can bound the the difference between $\mathcal{L}_{\mathcal{D}}^{\Gamma}(h)$ and $\mathcal{L}_{S}^{\Gamma}(h)$ for any given hypothesis $h \in \mathcal{H}$.

**Corollary 4.** *Assume that the loss function is upper bounded by $B \geq 0$. Then, for a fixed hypothesis $h \in \mathcal{H}$, $\delta \in (0,1)$, with probability at least $1 - \delta$, we have*

$$\mathcal{L}_{S}^{\Gamma}(h) \leq \mathcal{L}_{\mathcal{D}}^{\Gamma}(h) + B\|\Gamma\|_2\sqrt{\frac{\log\frac{2}{\delta}}{2}}.$$

*Proof of Corollary A.* Combining Theorem A with Lemma A and Corollary 4, we can show that for any $\delta \in (0,1)$, with probability at least $1 - \delta$, the followings hold:

$$\mathcal{L}_{\mathcal{D}_{\mathcal{T}}}(w_S) \leq \mathcal{L}_S^{\Gamma}(w_S) + \frac{\|\Gamma\|_\infty\rho^2R^2}{\lambda} + \left(\frac{\rho^2R^2(\|\Gamma\|_2^2+\|\Gamma\|_\infty)}{\lambda} + \|\Gamma\|_\infty B(\Gamma)\right)\sqrt{\frac{N\log\frac{1}{\delta}}{2}} + \|\Gamma^{\mathcal{S}}\|_1\,\mathrm{dist}_{\mathcal{Y}}(\mathcal{D}_{\mathcal{T}},\mathcal{D}_{\mathcal{S}})$$

$$\leq \mathcal{L}_S^{\Gamma}(w^*) + \frac{\|\Gamma\|_\infty\rho^2R^2}{\lambda} + \left(\frac{\rho^2R^2(\|\Gamma\|_2^2+\|\Gamma\|_\infty)}{\lambda} + \|\Gamma\|_\infty B(\Gamma)\right)\sqrt{\frac{N\log\frac{1}{\delta}}{2}} + \|\Gamma^{\mathcal{S}}\|_1\,\mathrm{dist}_{\mathcal{Y}}(\mathcal{D}_{\mathcal{T}},\mathcal{D}_{\mathcal{S}})$$

$$\leq \mathcal{L}_{\mathcal{D}}^{\Gamma}(w^*) + \|\Gamma\|_2B(\Gamma)\sqrt{\frac{\log\frac{4}{\delta}}{2}} + \frac{\|\Gamma\|_\infty\rho^2R^2}{\lambda}$$

$$+ \left(\frac{\rho^2R^2(\|\Gamma\|_2^2+\|\Gamma\|_\infty)}{\lambda} + \|\Gamma\|_\infty B(\Gamma)\right)\sqrt{\frac{N\log\frac{2}{\delta}}{2}} + \|\Gamma^{\mathcal{S}}\|_1\,\mathrm{dist}_{\mathcal{Y}}(\mathcal{D}_{\mathcal{T}},\mathcal{D}_{\mathcal{S}})$$

$$\leq \mathcal{L}_{\mathcal{D}_{\mathcal{T}}}(w^*) + \|\Gamma\|_2B(\Gamma)\sqrt{\frac{\log\frac{4}{\delta}}{2}} + \frac{\|\Gamma\|_\infty\rho^2R^2}{\lambda}$$

$$+ \left(\frac{\rho^2R^2(\|\Gamma\|_2^2+\|\Gamma\|_\infty)}{\lambda} + \|\Gamma\|_\infty B(\Gamma)\right)\sqrt{\frac{N\log\frac{2}{\delta}}{2}} + 2\|\Gamma^{\mathcal{S}}\|_1\,\mathrm{dist}_{\mathcal{Y}}(\mathcal{D}_{\mathcal{T}},\mathcal{D}_{\mathcal{S}}),$$

and

$$\mathcal{L}_{\mathcal{D}_{\mathcal{T}}}(w_S) \leq \mathcal{L}_S^{\Gamma}(w_S) + \frac{2\|\Gamma\|_\infty\|\Gamma\|_2\rho^2R^2}{\lambda}\sqrt{2N\log\frac{4}{\delta}} + \|\Gamma\|_2B(\Gamma)\sqrt{\frac{\log\frac{2}{\delta}}{2}} + \|\Gamma^{\mathcal{S}}\|_1\,\mathrm{dist}_{\mathcal{Y}}(\mathcal{D}_{\mathcal{T}},\mathcal{D}_{\mathcal{S}})$$

$$\leq \mathcal{L}_{\mathcal{D}}^{\Gamma}(w^*) + \frac{2\|\Gamma\|_\infty\|\Gamma\|_2\rho^2R^2}{\lambda}\sqrt{2N\log\frac{8}{\delta}} + 2\|\Gamma\|_2B(\Gamma)\sqrt{\frac{\log\frac{4}{\delta}}{2}} + \|\Gamma^{\mathcal{S}}\|_1\,\mathrm{dist}_{\mathcal{Y}}(\mathcal{D}_{\mathcal{T}},\mathcal{D}_{\mathcal{S}})$$

$$\leq \mathcal{L}_{\mathcal{D}_{\mathcal{T}}}(w^*) + \frac{2\|\Gamma\|_\infty\|\Gamma\|_2\rho^2R^2}{\lambda}\sqrt{2N\log\frac{8}{\delta}} + 2\|\Gamma\|_2B(\Gamma)\sqrt{\frac{\log\frac{4}{\delta}}{2}} + 2\|\Gamma^{\mathcal{S}}\|_1\,\mathrm{dist}_{\mathcal{Y}}(\mathcal{D}_{\mathcal{T}},\mathcal{D}_{\mathcal{S}}).$$

$\square$

### 3.5 Bound $\text{dist}_{\mathcal{Y}}(\mathcal{D}_{\mathcal{T}}, \mathcal{D}_{\mathcal{S}})$

Given a hypothesis class $\mathcal{H}$, the following propositions show that the discrepancy distance between a distribution $\mathcal{D}$ and its empirical distribution $\hat{\mathcal{D}}$ can be bounded in terms of the Rademacher complexity of $\mathcal{H}$.

**Proposition 1.** *Assume that the loss function is upper bounded by $B \geq 0$, and is $\rho$-Lipschitz. Let $\mathcal{D}$ be a distribution over $\mathcal{X} \times \mathcal{Y}$ and let $\hat{\mathcal{D}}$ be the corresponding empirical distribution for a sample $S = \{(x_i, y_i)\}_{i=1}^{N}$. Then, for any $\delta > 0$, with probability at least $1 - \delta$, the following holds:*

$$\text{dist}_{\mathcal{Y}}(\mathcal{D}, \hat{\mathcal{D}}) \leq 2\rho\Re_{\mathcal{D}}(\mathcal{H}) + 3B\sqrt{\frac{\log\frac{1}{\delta}}{2N}}$$

**Proposition 2.** *Let $\mathcal{H}$ be a hypothesis class mapping $\mathcal{X}$ to $\{-1, 1\}$, and let $\ell_{01}$ be the 0-1 loss.[4] Let $\mathcal{D}$ be a distribution over $\mathcal{X} \times \mathcal{Y}$ and let $\hat{\mathcal{D}}$ be the corresponding empirical distribution for a sample $S = \{(x_i, y_i)\}_{i=1}^{N}$. Then, for any $\delta > 0$, with probability at least $1 - \delta$, the following holds:*

$$\text{dist}_{\mathcal{Y}}(\mathcal{D}, \hat{\mathcal{D}}) \leq 4\Re_{\mathcal{D}}(\mathcal{H}) + 3\sqrt{\frac{\log\frac{1}{\delta}}{2N}}$$

The proof of Propositions 1 and 2 follows the standard techniques as described in [10], and therefore is omitted here.

*Proof of Lemma A.* Combining Propositions 1, 2, and Rademacher complexity bound of an argument stable algorithm (Theorem 1 in [7]), and by the triangle inequality property of $\mathcal{T}$-discrepancy, we immediately obtain Lemma A. $\qquad\square$

*Proof of Lemma B.* We show that for any hypothesis $h$, the following holds:

$$1 - \left( \frac{1}{N_{\mathcal{T}}} \sum_{x_i^{\mathcal{T}}:y_i^{\mathcal{T}}=1} \mathbb{1}_{h(x_i^{\mathcal{T}})=1} + \frac{1}{N_{\mathcal{S}}} \sum_{x_i^{\mathcal{S}}:y_i^{\mathcal{S}}=0} \mathbb{1}_{h(x_i^{\mathcal{S}})=1} + \frac{1}{N_{\mathcal{T}}} \sum_{x_i^{\mathcal{T}}:y_i^{\mathcal{T}}=0} \mathbb{1}_{h(x_i^{\mathcal{T}})=0} + \frac{1}{N_{\mathcal{S}}} \sum_{x_i^{\mathcal{S}}:y_i^{\mathcal{S}}=1} \mathbb{1}_{h(x_i^{\mathcal{S}})=0} \right)$$

$$= \frac{1}{2N_{\mathcal{T}}} \left( \sum_{x_i^{\mathcal{T}}:y_i^{\mathcal{T}}=1} \mathbb{1}_{h(x_i^{\mathcal{T}})=1} + \sum_{x_i^{\mathcal{T}}:y_i^{\mathcal{T}}=0} \mathbb{1}_{h(x_i^{\mathcal{T}})=1} + \sum_{x_i^{\mathcal{T}}:y_i^{\mathcal{T}}=1} \mathbb{1}_{h(x_i^{\mathcal{T}})=0} + \sum_{x_i^{\mathcal{T}}:y_i^{\mathcal{T}}=0} \mathbb{1}_{h(x_i^{\mathcal{T}})=0} \right)$$

$$+ \frac{1}{2N_{\mathcal{S}}} \left( \sum_{x_i^{\mathcal{S}}:y_i^{\mathcal{S}}=1} \mathbb{1}_{h(x_i^{\mathcal{S}})=1} + \sum_{x_i^{\mathcal{S}}:y_i^{\mathcal{S}}=0} \mathbb{1}_{h(x_i^{\mathcal{S}})=1} + \sum_{x_i^{\mathcal{S}}:y_i^{\mathcal{S}}=1} \mathbb{1}_{h(x_i^{\mathcal{S}})=0} + \sum_{x_i^{\mathcal{S}}:y_i^{\mathcal{S}}=0} \mathbb{1}_{h(x_i^{\mathcal{T}})=0} \right)$$

$$- \left( \frac{1}{N_{\mathcal{T}}} \sum_{x_i^{\mathcal{T}}:y_i^{\mathcal{T}}=1} \mathbb{1}_{h(x_i^{\mathcal{T}})=1} + \frac{1}{N_{\mathcal{S}}} \sum_{x_i^{\mathcal{S}}:y_i^{\mathcal{S}}=0} \mathbb{1}_{h(x_i^{\mathcal{S}})=1} + \frac{1}{N_{\mathcal{T}}} \sum_{x_i^{\mathcal{T}}:y_i^{\mathcal{T}}=0} \mathbb{1}_{h(x_i^{\mathcal{T}})=0} + \frac{1}{N_{\mathcal{S}}} \sum_{x_i^{\mathcal{S}}:y_i^{\mathcal{S}}=1} \mathbb{1}_{h(x_i^{\mathcal{S}})=0} \right)$$

$$= \frac{1}{2N_{\mathcal{T}}} \left( - \sum_{x_i^{\mathcal{T}}:y_i^{\mathcal{T}}=1} \mathbb{1}_{h(x_i^{\mathcal{T}})=1} + \sum_{x_i^{\mathcal{T}}:y_i^{\mathcal{T}}=0} \mathbb{1}_{h(x_i^{\mathcal{T}})=1} + \sum_{x_i^{\mathcal{T}}:y_i^{\mathcal{T}}=1} \mathbb{1}_{h(x_i^{\mathcal{T}})=0} - \sum_{x_i^{\mathcal{T}}:y_i^{\mathcal{T}}=0} \mathbb{1}_{h(x_i^{\mathcal{T}})=0} \right)$$

$$+ \frac{1}{2N_{\mathcal{S}}} \left( \sum_{x_i^{\mathcal{S}}:y_i^{\mathcal{S}}=1} \mathbb{1}_{h(x_i^{\mathcal{S}})=1} - \sum_{x_i^{\mathcal{S}}:y_i^{\mathcal{S}}=0} \mathbb{1}_{h(x_i^{\mathcal{S}})=1} - \sum_{x_i^{\mathcal{S}}:y_i^{\mathcal{S}}=1} \mathbb{1}_{h(x_i^{\mathcal{S}})=0} + \sum_{x_i^{\mathcal{S}}:y_i^{\mathcal{S}}=0} \mathbb{1}_{h(x_i^{\mathcal{T}})=0} \right)$$

$$= \frac{1}{2}(2\mathcal{L}_{\hat{\mathcal{D}}_{\mathcal{T}}}(h) - 1) + \frac{1}{2}(1 - 2\mathcal{L}_{\hat{\mathcal{D}}_{\mathcal{T}}}(h))$$

$$= \mathcal{L}_{\hat{\mathcal{D}}_{\mathcal{T}}}(h) - \mathcal{L}_{\hat{\mathcal{D}}_{\mathcal{S}}}(h)$$

$$\square$$

### 3.6 Proof of Proposition A

**Lemma 9.** *Let $\mathcal{H}$ be a hypothesis class of real-valued functions returned by the transfer learning algorithm (1) with a $\rho$-Lipschitz continuous loss function. The convex hull of $\mathcal{H}$ is defined as*

$$\mathcal{F} = \left\{ \sum_{k=1}^{K} \mu_k h_k(x) : \sum_{k=1}^{K} \mu_k = 1, \mu_k \geq 0, h_k \in \mathcal{H}, \forall k = \{1, \ldots, K\} \right\}.$$

*Define $\Gamma = [\Gamma_1, \ldots, \Gamma_K] \in \mathbb{R}^{N \times K}$, where for any $k \in 1, \ldots, K$, $\Gamma_k = [\Gamma_k^{\mathcal{T}}; \Gamma_k^{\mathcal{S}}] = [\gamma_{1,k}^{\mathcal{T}}, \ldots, \gamma_{N_{\mathcal{T}},k}^{\mathcal{T}}; \gamma_{1,k}^{\mathcal{S}}, \ldots, \gamma_{N_{\mathcal{S}},k}^{\mathcal{S}}]^{\top} \in \mathbb{R}^N$ are the weights for the $k$-th base learner. Then, for any $\delta \in (0,1)$, we probability at least $1 - \delta$, we have*

$$\Re_{\mathcal{D}_{\mathcal{T}}}(\mathcal{F}) \leq \frac{\gamma_{\infty}^{\mathcal{T}} \rho R^2}{\lambda} \sqrt{2N \log \frac{2}{\delta}}$$

*where $\gamma_{\infty}^{\mathcal{T}} = \max_k \{\|\Gamma_k^{\mathcal{T}}\|_{\infty}\}_{k=1}^{K}$ is the largest weight of the target sample over all the boosting iterations.*

*Proof.* We derive the generalization bound from the unweighted target training sample, treating source domain sample can be treated as a regularizer [9, 8]. Then, follow the similar proof schema as in Theorem 6.2 of [13], we have

$$\Re_{\mathcal{D}_{\mathcal{T}}}(\mathcal{F}) = \frac{1}{N_{\mathcal{T}}} \mathbb{E} \sup_{\substack{h_1 \in \mathcal{H}_1, \ldots, h_K \in \mathcal{H} \\ \mu_k \geq 0, \sum_{k=1}^{K} \mu_k \leq 1}} \sum_{i=1}^{N_{\mathcal{T}}} \sigma_i \sum_{k=1}^{K} \mu_k h_k(x_i)$$

$$= \frac{1}{N_{\mathcal{T}}} \mathbb{E} \sup_{\substack{h_1 \in \mathcal{H}_1, \ldots, h_K \in \mathcal{H} \\ \mu_k \geq 0, \sum_{k=1}^{K} \mu_k \leq 1}} \sum_{k=1}^{K} \mu_k \left( \sum_{i=1}^{N_{\mathcal{T}}} \sigma_i h_k(x_i) \right)$$

$$= \frac{1}{N_{\mathcal{T}}} \mathbb{E} \sup_{h_1 \in \mathcal{H}_1, \ldots, h_K \in \mathcal{H}} \max_{k \in \{1, \ldots, K\}} \sum_{i=1}^{N_{\mathcal{T}}} \sigma_i \langle h_k - \mathbb{E} h_{k,S}, x_i \rangle$$

$$\leq \frac{1}{N_{\mathcal{T}}} \mathbb{E} \max_{k \in \{1, \ldots, K\}} \sup_{h_1 \in \mathcal{H}_1, \ldots, h_K \in \mathcal{H}} \| h_k - \mathbb{E} h_{k,S} \| \left\| \sum_{i=1}^{N_{\mathcal{T}}} \sigma_i x_i \right\|_2 \qquad \text{Cauchy-Schwarz inequality}$$

$$\leq \frac{1}{N_{\mathcal{T}}} \max_{k \in \{1, \ldots, K\}} \left\{ \frac{\|\Gamma_k^{\mathcal{T}}\|_{\infty} \rho R}{\lambda} \sqrt{2N_{\mathcal{T}} \log \frac{2}{\delta}} \right\} \mathbb{E} \left\| \sum_{i=1}^{N_{\mathcal{T}}} \sigma_i \gamma_i^k x_i \right\|_2$$

$$\leq \frac{\gamma_{\infty}^{\mathcal{T}} \rho R}{N_{\mathcal{T}} \lambda} \sqrt{2N_{\mathcal{T}} \log \frac{2}{\delta}} R \sqrt{N_{\mathcal{T}}} = \frac{\gamma_{\infty}^{\mathcal{T}} \rho R^2}{\lambda} \sqrt{2 \log \frac{2}{\delta}}$$

Note that compared with Theorem 6.2 of [13], the main difference in our proof is that for each base learner, its hypothesis class defined by learning algorithm (1) is different from others. $\square$

*Proof of Proposition A.* Given Lemma 9, by following the standard proof schema as in [13], we immediately obtain Proposition A. $\square$

## 4 Additional Experimental Results

In this section, we report more detailed results on the 20 Newsgroup data set.

### 4.1 Learning Curves with Different Amount of Target Examples

We set the ratio of target samples as $[0.01, 0.02, 0.05, 0.1, 0.2, 0.3, 0.4, 0.5, 0.6, 0.7, 0.8]$, and the results are shown in Figure 1.

## 4.2 Sensitivity of the Parameters $\rho_{\mathcal{S}}$

We fixed $\rho_{\mathcal{T}} = 0$ and varied $\exp(\rho_{\mathcal{S}})$ in the range $[0.1, \ldots, 0.9]$. Figure 2 shows the results averaged over all transfer problems on the 20Newsgroups data set, showing that as the size of the target sample increases, the influence of the hyper-parameter on performance decreases.

Figure 1: Test error rates (%) with different sizes of target sample on different tasks and on average across all tasks. gapBoost

Figure 2: Test error rates (%) averaged across all tasks with respect to the values of the hyper-parameter $\rho_{\mathcal{S}}$ for varying sample sizes. Rightmost graphic shows results averaged over all sample sizes. Our algorithm, gapBoost becomes less sensitive to the choice of $\rho_{\mathcal{S}}$ as the target sample grows larger. In all cases, there is a range of $\rho_{\mathcal{S}}$ that outperforms all baselines. Error bars represent standard error.

## Footnotes

[1]The unlabeled target domain sample can have indirect effects on the learning problem through the instance weights $\Gamma^\mathcal{S}$, as they are usually learned from both the source and target domains [5, 4].

[2]We use numerical sequencing for intermediate results, and alphabetical sequencing for main results.

[3]We write $\ell(h(x), y)$ as $\ell_z(h)$ for simplicity.

[4]0-1 loss is not $\rho$-Lipschitz, but it can be upper bounded by hinge loss, which is 1-Lipschitz. Therefore, the generalization bound still holds by slightly modifying it.