[Reviews · NeurIPS 2019]

Reviewer 1



==================================== After rebuttal I thank the authors for their reply, they have managed to clarify some of my concerns and overall I vote for acceptance of the paper. ============= This paper introduces a boosting method for transfer learning with instance re-weighting in the setting where labeled data are available in both training and target tasks. Theorem 1 provides a bound for the population error on the target task, and motivates four instance re-weighting principles''. A practical procedure is introduced, which achieves competitive results on two standard datasets for transfer learning. Novelty: To my knowledge, the theoretical analysis carried out by the authors in the context of fully labeled data is novel. In particular, I am not aware of previous bounds including the performance gap from Def 1. I did not, however, go through the proofs in the Appendix in detail. Significance: I find the results interesting, and found the bound in 2) insightful, as well as the resulting algorithm. The empirical results are also strong, and present an advantage over competing methods (though small for TransferBoost). Clarity: The paper is overall well structured and the mains ideas are easily conveyed. However, I find that improving the presentation of gapBoost in Section 3.3 would benefit the paper. The algorithm is rather obscure at first glance, and the actual procedure is not explained in the main text (only a very high level view). Key elements, such as \rho_S and \rho_T, are not really defined, and lead to confusion on a first read since the notation overlaps with the Lipschitz constant. The procedure for re-weighting the new weights at each iteration based on the performance of the different learners is not explained in the text. Moreover, it is hard to understand how the four principles should be interpreted in Section 3.1, since they contradict each other (as the authors point out), and what the authors want to achieve is only made clear after reading the whole Section. I believe the trade-off could be made clearer, perhaps after the results from 3.2, and that the main algorithm could be described in more detail. In addition to the previous comments, I have the following concerns. 1) What can we expect when linearity does not hold? Linearity is assumed for the Theorem (line 115), and in the experiments (line 252). However, the method is presented as general, see for instance line 207. Have the authors experimented with this, and what can we expect from the theory? 2) An interesting aspect of the algorithm is the trade-off between different values of \rho_S and \rho_T. Is there a principled way to select them, and would it be possible to have them vary together with the instance weights? Why is \rho_T=0 in the experiments? It seems that at least a simple heuristic based on the number of examples available in each task could be derived. Finally, are \rho_S and \rho_T fixed as the ratio of training examples increases in Fig 1? 3) Often, we do have unlabeled data in the test set. How could we make use of these data in the proposed theory/algorithm? As a minor comment, why is the notation for the weights changed from \Gamma to D in the algorithm?

Reviewer 2



1. Originality. The authors propose a novel measure to evaluate domain discrepancy; the idea is interesting and novel. 2. Quality. The manuscript is well-written. 3. Clarity. The manuscript provides methods and detail analysis. However, as the experimental settings are semi-supervised based, and the baseline methods are a little out-of-date. I suggest the authors add more baselines to demonstrate the effectiveness of the proposed method. An example is listed as follows. Zhang L, Zuo W, Zhang D. LSDT: Latent sparse domain transfer learning for visual adaptation[J]. IEEE Transactions on Image Processing, 2016, 25(3): 1177-1191. ================ After rebuttal Although the baseline methods are all boosting-based, I think the performance is an important factor for a good manuscript. ================ 4. Significance. The quality is good, and the technical significance is high.

Reviewer 3



I would consider Theorem 1 and analysis behind it as a main contribution of the paper. Even though, the bound has too many "moving parts", the trade-off between relevant quantities seems to make sense. One caveat is L2 norm of the weights: when inequality ||Γ||_2 <= sqrt(N) ||Γ||_∞ is tight (for sufficiently large N), the bound of Theorem 1 appears to be vacuous (it is multiplied by the term of order sqrt(N)). Line 182: "...||Γ||_∞ << 1/N_T, which implies that transfer learning has a faster convergence rate than single-task learning" -- this is not entirely true, as the bound is also controlled by the discrepancy, which can be large enough for any source sample size. == post-rebuttal reply I agree with the authors (and raise the score): the rate seems to be fine (the crucial point here is that ||Γ||_2 is squared). One additional note: the bound seems to hold only for a fixed Γ. It should be okay for a general intuition, but to for the design of an algorithm which minimizes the bound simultaneously in h and Γ simultaneously, we should have a bound uniform in Γ (or at least a union bound). Authors could discuss this in a final version.

[Author Response · NeurIPS 2019]

We thank the reviewers for the detailed comments. We note that reviewers 2 and 3 think highly of the quality of the
paper. If our responses have addressed your concerns, we hope that you would give an accept recommendation.

**Reviewer 2**

**1.** Presentation of gapBoost: Thank you for the suggestion. We will re-organize Section 3.
**2.** Nonlinear extension: our analysis can be extended to (nonlinear) kernel models based on a reproducing kernel
Hilbert space. The theoretical analysis of general nonlinear models (e.g., deep nets) is challenging since their loss
landscape is usually non-convex. However, motivated by the empirical success of convex optimization methods for
fitting complex deep nets, we hypothesize that we could still leverage the intuition behind our gap minimization
principle to create novel deep transfer methods. In future work, we plan to empirically verify this conjecture.
**3.** In Section 3.5 of Appendix, we have shown that the $\mathcal{Y}$-discrepancy can be bounded from training data by
constructing a classification problem, which may be used as a guideline to select parameters in a principled way. We
choose $\rho_{\mathcal{T}} = 0$ as it corresponds to no punishment for the target data (the simplest setting). We have run additional
experiments by varying both parameters. In Fig. 1, we can observe that by properly choosing both parameters (e.g.,
$\rho_{\mathcal{T}} = \log 2, \rho_{\mathcal{S}} = 0$), we may obtain even better results. As you point out, we could use a simple heuristic like choosing
a relatively larger $\rho_{\mathcal{S}}$ when target data is small in order to leverage source data, as shown in Fig. 1(a). As the target data
increase, the results are less sensitive to the parameter. As long as $\rho_{\mathcal{T}} > \rho_{\mathcal{S}}$, the performance of gapBoost is stable
over a wide range of values of parameters, as shown in Fig. 1(b)–1(d). In Fig. 1 in the paper, we fixed $\rho_{\mathcal{T}} = 0$ and
$\rho_{\mathcal{S}} = \log \frac{1}{2}$. This will be made more explicit in the revised version.
**4.** There are various measures for unlabeled data proposed in the literature (see the references in Line 57), which
could be incorporated into our work. The notion of discrepancy [25] (the unsupervised version of $\mathcal{Y}$-discrepancy) is
particularly relevant, due to its consistency with the notion used in our paper. We will also be working on generalizing
the notion of gap to the unsupervised learning (domain adaptation) setting.

**Reviewer 3**

Thank you for your comments and pointing out the reference. We will add a qualitative comparison in our paper. Please
note that the current baselines methods are all boosting-based approaches in order to make a fair comparison.

**Reviewer 4**

**1.** Vacuous bound: The inequality $||\Gamma||_2 \leq \sqrt{N}||\Gamma||_\infty$ is tight when we assign equal weights to all data points. **Since** $\Gamma$
**is a probability simplex, we have** $||\Gamma||_2 = \frac{1}{\sqrt{N}}$ **and** $||\Gamma||_\infty = \frac{1}{N}$. **Then, after simplifying the multiplicative term**
$\sqrt{N}$, $\varepsilon_\Gamma$ **has a fast convergence rate of** $\mathcal{O}(\frac{1}{\sqrt{N}})$ **in this case,** which motivates Rule 2. In fact, we recover the learning
bound of assigning equal weights on source and target instances [3] (i.e., pooling-task approach). See also Remark 3 for
more discussions.
**2.** Moving parts: Thank you for noting that the trade-off between the multiple terms is intuitively reasonable, which
motivates the proposed rules.
**3.** Line 182: As you correctly point out, the bound is controlled by the discrepancy—it is also shown in the last term
of (2), which indeed motivates Rule 3. The convergence rate is in fact the convergence rate of $\varepsilon_\Gamma$. We will clarify this
point in the revised version.
**4.** Tools are straightforward ... largely inspired by [20]: While the tools are commonly used, we extend the existing
theoretical results in the following ways. First, we propose the novel notion of *performance gap*, revealing a new
principle for transfer learning. Second, we extend existing tools to their "weighted" version (e.g., weighted Rademacher
complexity/uniform stability/Hoeffding's inequality, see Appendix for details). Third, we develop the bounds for
$\mathcal{Y}$-discrepancy in the supervised learning context (the notion of discrepancy in [3], [25] is designed in the unsupervised
learning context). We also show that for 0-1 loss, the empirical $\mathcal{Y}$-discrepancy can be computed by constructing a
new classification problem. See Section 3.5 of Appendix for more details. Finally, we only use [20] to derive the
Rademacher bound after we have obtained the stability bound, and we extend it to our weighting setting.

Figure 1: Test error rates (%) with varying $\rho_{\mathcal{S}}$ and $\rho_{\mathcal{T}}$. The valley curves are obtained by setting $\rho_{\mathcal{T}} = 0$ (i.e., the purple curves in Fig. 2 of main paper). Hence the areas below the curve indicate better parameter configurations.

[Meta-Review · NeurIPS 2019]

The paper presents a novel theoretical analysis in the context of fully labeled data that is novel and sound. The methodological and algorithmic contributions based on a boosting strategy for a reweighed scheme is novel and shows good results. The experimental study can be improved with more baselines and datasets.